# Resolving the Tug-of-War: A Separation of Communication and Learning in Federated Learning

**Junyi Li**
Computer Science
University of Maryland College Park
junyili.ai@gmail.com

**Heng Huang** *
Computer Science
University of Maryland College Park
henghuanghh@gmail.com

## Abstract

Federated learning (FL) is a promising privacy-preserving machine learning paradigm over distributed data. In this paradigm, each client trains the parameter of a model locally and the server aggregates the parameter from clients periodically. Therefore, we perform the learning and communication over the same set of parameters. However, we find that learning and communication have fundamentally divergent requirements for parameter selection, akin to two opposite teams in a tug-of-war game. To mitigate this discrepancy, we introduce FedSep, a novel two-layer federated learning framework. FedSep consists of separated communication and learning layers for each client and the two layers are connected through decode/encode operations. In particular, the decoding operation is formulated as a minimization problem. We view FedSep as a federated bilevel optimization problem and propose an efficient algorithm to solve it. Theoretically, we demonstrate that its convergence matches that of the standard FL algorithms. The separation of communication and learning in FedSep offers innovative solutions to various challenging problems in FL, such as Communication-Efficient FL and Heterogeneous-Model FL. Empirical validation shows the superior performance of FedSep over various baselines in these tasks.

## 1 Introduction

In Federated Learning (FL) [37], a set of clients jointly solve a machine learning task under the coordination of a central server. The process of FL involves two category of operations: Learning and Communication. For the learning operation, each client optimizes their local objectives, and for the communication operation, clients exchange local parameters to facilitate knowledge sharing. In the existing FL pipeline, both Learning and Communication operations hinge on the same set of parameters. However, these two operations have fundamentally divergent requirements, akin to two teams engaged in a tug-of-war, each pulling in opposite directions. In fact, on the communication side, it is imperative that the parameters of all clients reside in a uniform space. Moreover, to mitigate the high communication costs, a major bottleneck in FL, it is beneficial to maintain these parameters within a low-dimensional space. In contrast, on the learning side, given the heterogeneity of devices, including variations in hardware and data distribution, it is advantageous to allow the parameter space to differ across clients. Furthermore, to accommodate the implementation of state-of-the-art large-scale machine learning models (such as Transformers [47]), a high-dimensional parameter space is desirable. In summary, a huge discrepancy of requirements to parameter selection exists between the communication and the learning in FL.

---

*This work was partially supported by NSF IIS 1838627, 1837956, 1956002, 2211492, CNS 2213701, CCF 2217003, DBI 2225775.

37th Conference on Neural Information Processing Systems (NeurIPS 2023).

To mitigate this discrepancy, we opt to 'break the rope' in this tug-of-war scenario. More precisely, we propose a two-layer structure on clients: a communication layer and a learning layer. As shown in Figure 1, we denote this framework as **FedSep**: The communication layer connects with the server and uploads/downloads parameter, while the learning layer performs local objective optimization. Then the communication layer and the learning layer are connected through encode/decode operations. The decode operation maps the parameter of the communication layer to the parameter of the learning layer and the encode operation performs the mapping in the opposite direction. More specifically, we formulate the decode operation as solving a minimization problem and does not assume an explicit relation between the parameter of communication layer and the parameter of the learning layer. In summary, we aim to use two distinct sets of parameters in FedSep to resolve the 'competition' between the communication and learning operations, meanwhile, the two set of parameters are close connected through decode/encode operations. Mathematically, FedSep can be formulated as solving a federated bilevel problem. The upper level problem corresponds to the learning problems on clients, while the lower level problem is the minimization problem in the decode operation. We propose an efficient algorithm to solve this bilevel problem, which is composed of three consecutive steps: Decode the communication parameter, Learning on local problems and Encode the learning parameter to the communication parameter. The convergence of the algorithm is guaranteed.

To demonstrate the flexibility of FedSep in incorporating the contradictory requirements of the communication and learning, we study two real-world FL tasks. In the first task, we study the communication-efficient FL task. In this task, we set the dimension of the communication parameter to be much smaller than the learning parameter, furthermore, we decode the communication parameter through a LASSO problem. In the second task, we study the model-heterogeneous FL task. In this task, the learning parameters on different clients have different dimension. We set the communication parameter to be the parameter of the server model, and the learning parameter be a subset of the communication parameter (server model). In particular, the decode operation is to select the subset adaptively based on the client's local data. We empirically verify the superior performance of our methods compared to other baselines. Finally, we summarize the main **contributions** of our work as follows:

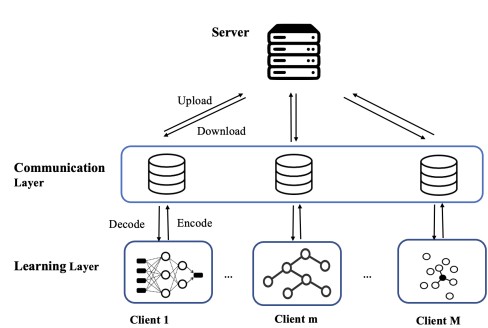

Figure 1: The structure of FedSep. FedSep follows the one-server-multiple-clients structure, in particular, each client has two layers: the communication layer and the learning layer. The two layers are connected through the decode and encode operations.

1. We propose a novel two-layer Federated Learning framework, *i.e.* **FedSep**, where the communication layer and the learning layer are separated on clients;
2. We let the decode operation from the communication layer to the learning layer be solving a minimization problem, and therefore, the **FedSep** framework has a bilevel optimization formulation. We propose an efficient algorithm to solve the bilevel problem and show that its convergence rate matches that of the standard Federated Learning algorithms;
3. We apply the **FedSep** framework to solve two important FL tasks: Communication-Efficient FL and Model-Heterogeneous FL. Numerical experiments show the superior performance of the methods based on FedSep.

**Notations.** $\nabla$ denotes the full gradient, $\nabla_x$ is the partial derivative for variable x, and higher-order derivatives follow similar rules. $||\cdot||_2$ is $\ell_2$-norm for vectors and the spectral norm for matrices. $AB$ denotes the multiplication of the matrix between the matrix $A$ and $B$. $[K]$ represents the sequence of integers from 1 to $K$.

## 2 Related Works

**Federated Learning.** FL is a novel paradigm for performing machine learning tasks over distributed data, but it also poses challenges such as heterogeneity [24], privacy [38] and communication

bottleneck [48]. In particular, communication cost is one of the major bottlenecks in FL. A widely used approach to reduce communication burden is through compression. Various compressors [48, 35, 45, 25] have been studied in the literature. Next, model-heterogeneous FL is also studied to solve the device heterogeneity issue. There are two main categories of heterogeneous model methods: knowledge-distillation (KD) [34, 22, 9] based and partial-training (PT) [11, 6, 1] based. KD-based methods treat the server model as the teacher and the clients' model as the student. In contrast, PT-based methods select a subset of the server model for each client, and in the aggregation phase, the server simply aggregates the updated sub-models.

**Bilevel Optimization.** Bilevel problem is a type of two-layer optimization problem [49]. In the machine learning community, Hyper-parameter Optimization problems [28, 7, 4, 12] are one of the mostly studied bilevel problems. Early bilevel optimization algorithms solved the lower problem exactly to update the upper variable. Recently, researchers developed algorithms that solve the lower problem approximately with a fixed number of steps and use the back-propagation technique to compute the approximate hyper-gradient [13, 36, 14, 40, 43]. More recently, single-loop algorithms [16, 18, 23, 30, 26, 8, 51, 31, 10, 20, 21] based on alternative updates of lower and upper level problems are proposed. However, less work studies bilevel problems under the federated setting. [46, 33, 32] studies a type of federated bilevel problems where the lower level problems are federated. In particular, [33] solves the noisy label problem in FL. Note that our FedSep is formulated as a type of federated bilevel problems, however, our problem is different from that in [46, 33]. Firstly, our problem has a unique lower problem for each client, moreover, FedSep impose unique constraints not covered by the existing methods. Please refer to Section B in Appendix for more related works.

# 3 Separating Communication and Learning in Federated Learning

In this section, we propose **FedSep**, a novel two-layer Federated Learning Framework. As shown in Figure 1, FedSep has the one-server-multiple-clients structure as in the classical Federated learning framework, we assume $M$ clients in the training. The difference between the FedSep and classical FL framework lies in the client. Different from the classical FL, FedSep includes two separated layers on clients: a communication layer and a learning layer. The two layers have different responsibilities. As the name shows, the learning layer is responsible of finishing the learning task, *e.g.* fitting a neural network over a dataset. We assume that the learning layer on client $m \in [M]$ has parameter $y^{(m)} \in \mathbb{R}^{d^{(m)}}$. Note that the choice of $y^{(m)}$ should be adapted to the specific setting of each client, such as the data distribution, hardware resources *etc.*. Next, the communication layer performs communication with the server (other clients), which includes the **upload** and **download** operations. We assume the communication layer on client $m$ has parameter $x \in \mathbb{R}^p$. Note that the parameter of communication layer has identical formulation for all clients. Finally, the communication layer and the learning layer are connected through the **encode** and **decode** operations.

To model the various potential relationship between the communication and learning layer, we abstract the decode operation as solving a minimization problem, and the encode operation be its reverse operation. As a result, we have the following bilevel formulation for the FedSep framework.

$$\min_{x \in \mathbb{R}^p} h(x) := \frac{1}{M} \sum_{m=1}^{M} h^{(m)}(x) := \frac{1}{M} \sum_{m=1}^{M} f^{(m)}(y_x^{(m)}), \quad y_x^{(m)} = \underset{y^{(m)} \in \mathbb{R}^{d^{(m)}}}{\arg\min} \, g^{(m)}(x, y^{(m)}) \quad (1)$$

In Eq. (1), $M$ is the number of clients; $f^{(m)}(y), m \in [M]$ denotes the local problem solved by the client $m$; $g^{(m)}(x, y)$ is the minimization problem solved in the decode operation by the client $m$; $x$ is the parameter of the communication layer; $y^{(m)}$ denotes the parameter of the learning layer on the client $m$.

Next, we propose a three-stage algorithm to solve Eq. (1). As shown by Algorithm 1, we perform $T$ global steps, and at each step $t \in T$, we randomly choose a subset of clients $\mathcal{M}_t$ to perform the training and then the server performs aggregation. The **Client training** is divided into three stages: **Decode stage** (lines 7-12), **Learning stage** (lines 14-17), and **Encode Stage** (line 19).

**Decode Stage.** In Eq. (1), the decode operation solves the lower optimization problem $g^{(m)}(x, y)$ to obtain $y_x^{(m)}$. We denote the decode operator on the $m_{th}$ client by $Dec^{(m)}\{\cdot\}$, thus we have: $y_x^{(m)} = Dec^{(m)}\{x\}$. In Algorithm 1, we solve $g^{(m)}(x, y)$ approximately with $I_{dec}$ steps of stochastic gradient

---

**Algorithm 1** Separating Communication and Learning in FL (**FedSep**)

---
1: **Input:** Initial states $x_1$; learning rates $\gamma, \eta$; mini-batchsize $b_x, b_y$
2: **for** $t = 1$ **to** $T$ **do**
3:     Randomly sample a subset $\mathcal{M}_t$ of clients;
4:     **for** $m \in \mathcal{M}_t$ in parallel **do**
5:         // Decode stage, estimate $y_x^{(m)} = Dec^{(m)}\{x\}$;
6:         Receive the global state $x_t$ from the server and set $x^{(m)} = x_t$, and initialize $y_0^{(m)}$;
7:         **for** $i = 1$ to $I_{dec}$ **do**
8:             Randomly sample a minibatch of $b_y$ samples $\mathcal{B}_y$;
9:             $y_i^{(m)} = y_{i-1}^{(m)} - \gamma\nabla_y g^{(m)}(x^{(m)}, y_{i-1}^{(m)}, \mathcal{B}_y)$
10:         **end for**
11:         Set $\hat{y}_0^{(m)} = y_{I_{dec}}^{(m)}$ as the estimation of $Dec^{(m)}(x^{(m)})$;
12:         // Learning stage, optimize $f^{(m)}(y)$;
13:         **for** $i = 1$ to $I$ **do**
14:             Randomly sample a minibatch of $b_y$ samples $\mathcal{B}_{\hat{y}}$;
15:             $\hat{y}_i^{(m)} = \hat{y}_{i-1}^{(m)} - \eta\nabla f^{(m)}(\hat{y}_{i-1}^{(m)}; \mathcal{B}_{\hat{y}})$;
16:         **end for**
17:         // Encode stage, encode the update of the learning layer back to the communication layer;
18:         Set $\Delta\hat{y}^{(m)} = \hat{y}_I^{(m)} - \hat{y}_0^{(m)}$ and compute $\Delta\hat{x}_t^{(m)} = \widetilde{Enc}^{(m)}\{\Delta\hat{y}^{(m)}\}$, where $\widetilde{Enc}^{(m)}\{\cdot\}$ is defined in Eq. (3) and we choose $|\mathcal{B}_x| = b_x$.
19:     **end for**
20:     $x_{t+1} = x_t - \frac{1}{|\mathcal{M}_t|}\sum_{m\in\mathcal{M}_t}\eta_g\Delta\hat{x}_t^{(m)}$
21: **end for**

---

descent (Line 8-11) and use the output as an approximation of $y_x^{(m)}$, *i.e..* $y_{I_{dec}}^{(m)} = \widetilde{Dec}^{(m)}(x^{(m)}) \approx y_x^{(m)}$, where $\widetilde{Dec}^{(m)}(\cdot)$ represents the approximation of the exact decoder $Dec^{(m)}\{\cdot\}$.

**Learning Stage.** Next, we perform $I$ steps of stochastic gradient descent to solve the local learning problem $f^{(m)}(y)$ (Line 14 - 17). This stage is similar to the local gradients in classical FL framework. Note we get $\hat{y}_I^{(m)}$ as the updated learning parameter $y^{(m)}$, more formally, we use $\Delta\hat{y}^{(m)} = \hat{y}_I^{(m)} - \hat{y}_0^{(m)} = \hat{y}_I^{(m)} - y_{I_{dec}}^{(m)}$ to represent the update in the learning stage.

**Encode Stage.** After the learning stage, we get the update of the local learning problem $\Delta\hat{y}^{(m)}$. Suppose we denote the encode operator $Enc^{(m)}\{\cdot\}$, then we get the update of the communication parameter as $\Delta\hat{x}^{(m)} = Enc^{(m)}(\Delta\hat{y}^{(m)})$ (Line 19). In particular, we choose the following encode operator:

$$Enc^{(m)}\{\cdot\} := -\nabla_x y_x^{(m)} = \nabla_{xy} g^{(m)}(x, y_x^{(m)})\big(\nabla_{yy} g^{(m)}(x, y_x^{(m)})\big)^{-1} \tag{2}$$

where $y_x^{(m)} = Dec^{(m)}(x)$ is the output of the exact decode operation and the second equality can be derived following the implicit function theorem under mild assumptions [16]. To reduce the computation complexity, we use Neumann series to approximate the inverse operation in Eq. (2). More specifically, we have the following approximation: $\big(\nabla_{yy} g^{(m)}(x, y)\big)^{-1} \approx \tau\sum_{q=0}^{Q}(I - \tau\nabla_{yy} g^{(m)}(x, y))^q$, where $Q$ and $\tau$ are some constants. Since in the decode stage, we use $y_x^{(m)} \approx y_{I_{dec}}^{(m)}$, we get a stochastic approximation of Eq. (2) as:

$$\widetilde{Enc}^{(m)}\{\cdot\} := \sum_{q=0}^{Q}\tau\nabla_{xy} g^{(m)}(x, y_{I_{dec}}^{(m)}; \mathcal{B}_x)(I - \tau\nabla_{yy} g^{(m)}(x, y_{I_{dec}}^{(m)}; \mathcal{B}_x))^q \tag{3}$$

*Remark* 1. We use Eq. (2) as the encode operator due to the following fact about the hyper-gradient:

$$\nabla h^{(m)}(x) = \nabla_x y_x^{(m)}(\nabla f^{(m)}(y_x^{(m)}))$$

Note that suppose $\Delta\hat{y}^{(m)} = -\nabla f^{(m)}(y_x^{(m)})$, which means the learning layer performs one step of gradient descent with learning rate 1 in the learning stage of Algorithm 1, then we have: $\nabla h^{(m)}(x) =$

$Enc\{\Delta\hat{y}^{(m)}\} = \Delta x^{(m)}$. Therefore, we update the communication parameter $x$ such that it optimize the overall problem $\nabla h^{(m)}(x)$.

**Server Aggregation.** Finally, the server aggregates the local updates of communication parameter $\Delta\hat{x}^{(m)}, m \in \mathcal{M}_t$ to get the new communication parameter as shown in Line 21 of Algorithm 1.

**Difference with existing bilevel algorithms.** Bilevel optimization problems are widely studied in the literature [16, 23]. However, FedSep cannot directly apply existing algorithms. In a standard bilevel algorithm, each step of update to the upper level variable requires (approximately) solving of the corresponding lower level problem. However, in each epoch of FedSep, we only solve the lower level problem once and perform multiple steps of update to the upper level variable.

## 3.1 Convergence Analysis

In this section, we provide the convergence analysis of Algorithm 1. Before we state the convergence result, we first make the following assumptions.

**Assumption 3.1.** Function $f^{(m)}(y)$ is possibly non-convex and $g^{(m)}(x, y)$ is $\mu$-strongly convex *w.r.t* $y$ for any given $x$.

**Assumption 3.2.** Function $f^{(m)}(y)$ is $L$-smooth and has $C_f$-bounded gradient.

**Assumption 3.3.** Function $g^{(m)}(x, y)$ is $L$-smooth; $\nabla_{xy}g^{(m)}(x, y)$ and $\nabla_{y^2}g^{(m)}(x, y)$ are Lipschitz continuous with constants $L_{xy}$ and $L_{y^2}$, respectively.

**Assumption 3.4.** We have unbiased stochastic first-order and second-order gradient oracle with bounded variance.

Note that Assumption 3.1-Assumption 3.3 are standard assumptions used to analyze bilevel problems [23, 16]. Assumption 3.4 is standard in analyzing stochastic optimization problems. For a full version of Assumption 3.4, please refer to Assumption C.1 in the Appendix.

In the learning stage of Algorithm 1, we perform $I$ steps of updates to optimize the learning problem, and the update has the form of: $\Delta\hat{y}^{(m)} = \hat{y}_I^{(m)} - \hat{y}_0^{(m)} = \sum_{i=0}^{I-1} -\eta\nabla f^{(m)}(\hat{y}_i^{(m)}; \mathcal{B}_{\hat{y}})$, therefore we have: $\Delta\hat{x}_t^{(m)} = \widetilde{Enc}^{(m)}\{\Delta\hat{y}^{(m)}\} = \sum_{i=0}^{I-1} \eta\widetilde{Enc}^{(m)}\{-\nabla f^{(m)}(\hat{y}_i^{(m)}; \mathcal{B}_{\hat{y}})\}$. We denote $\Delta\hat{x}_{t,i}^{(m)} = \widetilde{Enc}^{(m)}\{\nabla f^{(m)}(\hat{y}_i^{(m)}; \mathcal{B}_{\hat{y}})\}$. **Then the main effort of the proof is bounding the estimation error of $\Delta\hat{x}_{t,i}^{(m)}$ to the hyper-gradient $\nabla h^{(m)}(x)$.**

More specifically, we first show that $\Delta\hat{x}_{t,i}^{(m)}$ estimates a term $\mu_i^{(m)}$ with bounded variance and bias as stated in the following proposition:

**Proposition 3.5.** *Suppose Assumptions 3.1-3.3 and 3.4 hold and $\tau < \frac{1}{L}$, we have $\|\mathbb{E}_\xi[\Delta\hat{x}_{t,i}^{(m)}] - \mu_i^{(m)}\| \leq G_1$, where $\mu_i^{(m)} = -\nabla_{xy}g^{(m)}(x, y_{I_{dec}}^{(m)})(\nabla_{yy}g^{(m)}(x, y_{I_{dec}}^{(m)}))^{-1}\nabla f^{(m)}(\hat{y}_i^{(m)})$, and $\mathbb{E}\|\Delta\hat{x}_{t,i}^{(m)} - \mathbb{E}_\xi[\Delta\hat{x}_{t,i}^{(m)}]\|^2 \leq G_2^2$. $G_1$ and $G_2$ are some constants related to $Q$ in Eq. (3) and mini-batch size.*

Please refer to Proposition C.3 for the specific form of the constants in Proposition 3.5. Next, we have that $\mu_i^{(m)} \approx \nabla h^{(m)}(x)$ as shown in the following proposition:

**Proposition 3.6.** *Suppose Assumptions 3.2 and 3.3 hold, the following statements hold:*

   a) *$\|\mu_i^{(m)} - \nabla h^{(m)}(x)\|^2 \leq 2\hat{L}^2\|y_{I_{dec}}^{(m)} - y_x^{(m)}\|^2 + 2\kappa^2 L^2\|\hat{y}_i^{(m)} - y_x^{(m)}\|^2$, where $\hat{L} = O(\kappa^2)$.*

   b) *$h^{(m)}(x)$ is Lipschitz continuous in $x$ with constant $\bar{L}$ i.e., for any given $x_1, x_2 \in \mathbb{R}^p$, we have $\|\nabla h^{(m)}(x_2) - \nabla h^{(m)}(x_1)\|^2 \leq \bar{L}^2\|x_2 - x_1\|$, where $\bar{L} = O(\kappa^3)$.*

*where we denote the condition number as $\kappa = L/\mu$.*

Please refer to Proposition C.2 for the proof. Proposition 3.6 shows that $\mu_i^{(m)}$ estimates the true hyper-gradient $\nabla h^{(m)}(x)$ with two types of errors: the estimation error of the decode operation ($\|y_{I_{dec}}^{(m)} - y_x^{(m)}\|^2$) and the drift of the learning process ($\|\hat{y}_i^{(m)} - y_x^{(m)}\|^2$), where Lemma C.4 and

Lemma C.5 show bounds for these two errors. Furthermore, Proposition 3.6.b) shows that $h^{(m)}(x)$ is smooth. Next, we are ready to show the convergence of Algorithm 1:

**Theorem 3.7.** *Suppose Assumptions 3.1-3.4 hold, and we run $T$ iterations of Algorithm 1, with the learning rates satisfy $\gamma < \frac{1}{L}$, $\eta\eta_g < \frac{1}{2IL}$, then we have:*

$$\frac{1}{T}\sum_{t=1}^{T}\left(\mathbb{E}\|\nabla h(x_t)\|^2 + \frac{1}{2I}\sum_{i=1}^{I}\mathbb{E}\|\mathbb{E}_\xi[\bar{\Delta}\hat{x}_{t,i}]\|^2\right)$$

$$\leq \frac{2h(x_1)}{TI\eta\eta_g} + \frac{\eta\eta_g\bar{L}G_2^2}{b_xM} + 12I^2\kappa^2L^2\eta^2C_f + \frac{4I^2\kappa^2L^2\eta^2\sigma^2}{b_y}$$

$$+ \frac{4(3\kappa^2L^2 + \hat{L}^2)\gamma\sigma^2}{\mu b_y} + 2G_1^2 + 2(3\kappa^2L^2 + \hat{L}^2)(1-\mu\gamma)^{I_{dec}}C_0$$

*where the expectation $\mathbb{E}_\xi$ is* w.r.t *the noise of stochastic gradient. $b_x$, $b_y$ denotes the batch size, and $G_1$, $G_2$ and $C_0$ are some constants.*

Please refer to Theorem C.8 in the Appendix for the proof. We can further control the noise terms in Theorem 3.7 by choosing the learning rates carefully:

**Corollary 3.8.** *Suppose we choose the learning rates as $\gamma = \min(\frac{1}{2L}, (\frac{1}{C_\gamma T})^{1/2})$, $\eta = \min\left(1, \left(\frac{8Ib_xM\bar{L}h(x_1)}{TG_2^2}\right)^{1/2}, \left(\frac{4\bar{L}h(x_1)}{C_\eta I^2T}\right)^{1/3}\right)$ and $\eta_g = \frac{1}{2I\bar{L}}$, then we have:*

$$\frac{1}{T}\sum_{t=1}^{T}\mathbb{E}\left(\|\nabla h(x_t)\|^2 + \frac{1}{2I}\sum_{i=1}^{I}\|\mathbb{E}_\xi[\bar{\Delta}\hat{x}_{t,i}]\|^2\right) = O\left(\frac{\kappa^3}{T} + \left(\frac{\kappa^5}{T}\right)^{1/2} + \left(\frac{\kappa^6}{T^2}\right)^{1/3} + \tilde{G}\right)$$

*where $\tilde{G} = \kappa^2(1-\tau\mu)^{2(Q+1)} + \kappa^4(1-\mu\gamma)^{I_{dec}}$, $C_\eta$ and $C_\gamma$ are some constants.*

To reach an $\epsilon$ stationary point, we need to run Algorithm 1 with $T = O(\kappa^5\epsilon^{-2})$ number of iterations, furthermore, we need $Q = O(\kappa\log(\frac{\kappa}{\epsilon}))$, $I_{dec} = O(\kappa\log(\frac{\kappa}{\epsilon}))$. Note that this matches the iteration complexity of standard FL algorithms [50] up to some logarithm factors ($Q$ and $I_{dec}$). Note that Corollary 3.8 shows that the following term converges to 0: $\mathbb{E}\|\nabla h(x_t)\|^2 + \frac{1}{2I}\sum_{i=1}^{I}\mathbb{E}\|\mathbb{E}_\xi[\bar{\Delta}\hat{x}_{t,i}]\|^2$. In fact, for the first term, it shows that $x_t$ reaches to a stationary point of the bilevel problem $h(x)$, meanwhile, since we have:

$$\mathbb{E}_\xi[\bar{\Delta}\hat{x}_{t,i}] = \frac{1}{M}\sum_{m=1}^{M}\sum_{q=0}^{Q}\tau\nabla_{xy}g^{(m)}(x, y_{I_{dec}}^{(m)}) \times (I - \tau\nabla_{yy}g^{(m)}(x, y_{I_{dec}}^{(m)}))^q\nabla f^{(m)}(\hat{y}_i^{(m)})$$

Therefore, if $\|\nabla_{xy}g^{(m)}(x, y)\|$ is lower-bounded and $\|\nabla_{yy}g^{(m)}(x, y)\|$ is upper-bounded, we also get the learning parameter $\hat{y}_i^{(m)}$ converges to the stationary point of the local learning problem $f^{(m)}(y)$.

# 4 Application of FedSep to Real-world FL Problems

In FedSep, the separation of the communication layer and the learning layer makes it able to solve various challenges in Federated Learning. In this section, we apply FedSep to solve two challenging problems in Federated Learning: communication-efficient FL and model-heterogeneous FL.

## 4.1 Communication-efficient Federated Learning

Communication-cost is one of the major bottlenecks in Federated Learning due to slow connections between clients and the server. In Federated Learning, compression is commonly employed to mitigate communication costs, where clients compress the local updates before transferring to the server. A common compressor is choosing the Top-K coordinates, however, this approach is only good at reducing the upload communication cost. Since clients have different Top-K coordinates, the aggregated updates are often dense. More complicated approaches can save both upload and download communication cost, such as the Count-sketch based compressor [42]. In fact, we can develop a simple yet effective approach to reduce the communication cost based on FedSep.

In our FedSep framework, suppose we have the learning parameter $\theta \in \mathbb{R}^d$, and the communication parameter $\omega \in \mathbb{R}^p$. We choose $p \ll d$ to have a communication efficient federated learning algorithm. More specifically, we consider the following formulation:

$$\min_{\omega \in \mathbb{R}^p} \frac{1}{M} \sum_{m=1}^{M} \mathcal{L}(\theta_\omega^{(m)}; \mathcal{D}_{tr}^{(m)}) \ s.t. \ \theta_\omega^{(m)} = \arg\min_{\theta \in \mathbb{R}^d} \frac{1}{2} \left\| S^{(m)}\theta - \omega \right\|_2^2 + \beta \left\| \theta \right\|_1 \quad (4)$$

where $\mathcal{D}_{tr}^{(m)}$ denotes the training distribution of the $m_{th}$ client; $\mathcal{L}(\cdot)$ denotes the loss function. In particular, $S^{(m)} \in \mathbb{R}^{p \times d}$ ($p \ll d$) is a random sketch matrix whose coordinates are sampled from Gaussian distribution. We choose the quadratic optimization problem (LASSO) as the decode function.

Eq. (4) is a special case of Eq. (1): $\theta$ corresponds to the learning parameter $y$, $\omega$ corresponds to the communication parameter $x$, $\mathcal{L}(\theta; \mathcal{D}_{tr}^{(m)})$ corresponds to the learning problem $f^{(m)}(y)$ and the decoding problem is a LASSO problem. We can solve it through Algorithm 1.

The LASSO problem of the decode operation involves a non-smooth $L_1$ regularization term and we solve it with the standard proximal gradient method [3]. For the encode operation, we have:

$$Enc^{(m)}\{\cdot\} = -\nabla_\omega \theta_\omega^{(m)} = \gamma S^{(m)} U \left( I - \left( I - \gamma (S^{(m)})^T S^{(m)} \right) U \right)^{-1} \quad (5)$$

where $I$ is the identity matrix and $U = Diag\{\mathbb{I}_{\gamma\beta}(\theta_\omega^{(m)} + \gamma(S^{(m)})^T(\omega - S^{(m)}\theta_\omega^{(m)}))\}$ where $\mathbb{I}_{\gamma\beta}(\cdot)$ is an indicator operator, which outputs 1 if the absolute value of the input is greater than $\gamma\beta$. Please refer to Appendix A.1 for more details of the proximal gradient method and the encode operator.

*Remark* 2. If we set $\beta = 0$ in Eq. (4), *i.e.* we remove the sparsity constraints, both encode and decode operators have explicit solutions. More specifically, we have the decode operator $Dec^{(m)}\{\cdot\} := \left( (S^{(m)})^T S^{(m)} \right)^{-1} (S^{(m)})^T$ and the encode operator $Enc^{(m)}\{\cdot\} := S^{(m)} \left( (S^{(m)})^T S^{(m)} \right)^{-1}$. If we choose $S^{(m)} = S$ for all $m \in [M]$, we get a linear compressor.

## 4.2 Model-Heterogeneous Federated Learning

In practical Federated Learning applications, the scale of the model is inherently limited by the on-device resources of the participating clients, which often exhibit significant diversity. As a result, we can choose different scale of models based on the available resources of each individual client. One widely-used approach is the sub-model extraction, *i.e.* each client selects a part of the server model to perform local training. Different strategies can be used to do extraction: fixed [11, 2], random [6] and roll [1]. In contrast, we provide a data-dependent approach to select the sub-models. As shown by the following formulation:

$$\min_{\omega \in \mathbb{R}^p} \frac{1}{M} \sum_{m=1}^{M} \mathcal{L}(\theta_\omega^{(m)}; \mathcal{D}_{tr}^{(m)}) \quad (6)$$

$$s.t. \ \theta_\omega^{(m)} = a_\omega^{(m)} \odot \omega, \ a_\omega^{(m)} = \arg\min_{a \in \{0,1\}^p} \mathcal{L}(a \odot \omega; \mathcal{D}_{val}^{(m)}) + \beta \mathcal{R}(T(a), p^{(m)} T_{tol})$$

where $\mathcal{D}_{tr}^{(m)}$ denotes the training distribution; $\mathcal{L}(\cdot)$ denotes the loss function and $\theta$ and $\omega$ are the learning parameter. Note that $\omega$ denotes the parameter of the full model and $\theta$ is the parameter of the sub-model. In particular, we denote a mask vector $a_\omega^{(m)}$, whose value is from $\{0, 1\}$. $\odot$ represents the coordinate-wise multiplication.

For the decode operation, we have $\theta_\omega^{(m)} = a_\omega^{(m)} \odot \omega$. So we first need to find an optimal mask $a_\omega^{(m)}$, instead of solving the complicated integer programming as in Eq. (6), we solve the following relaxed continuous problem:

$$a_\omega^{(m)} = \arg\min_{a \in [0,1]^p} \mathcal{L}(a \odot \omega; \mathcal{D}_{val}^{(m)}) + \beta \mathcal{R}(T(a), p^{(m)} T_{tol}) \quad (7)$$

Eq. (7) includes two parts. $\mathcal{L}(a \odot \omega; \mathcal{D}_{val}^{(m)})$ measures the loss of the extracted sub-model over $\mathcal{D}_{val}^{(m)}$. The second part $\mathcal{R}(T(a), p^{(m)} T_{tol})$ is called resource loss [15]. $T_{tol}$ denotes FLOPS of

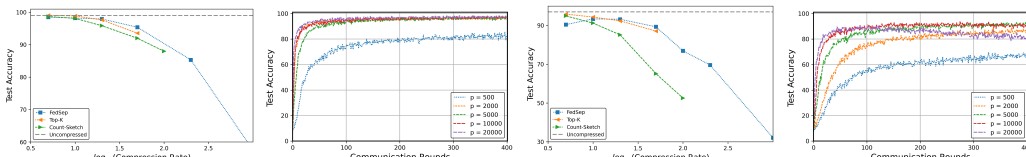

Figure 2: Test Accuracy w.r.t Communication Rate for FedSep and other baseline methods for MNIST Dataset. The first two plots show results under the I.I.D case, and the last two plots show results for the Non-I.I.D case. $p$ in the second and the fourth plot is the dimension of the communication parameter. The local learning steps are set as $I = 5$.

the server model and $T(a)$ denotes FLOPS of the sub-model. Therefore, $p^{(m)} \in [0, 1]$ controls the size of the sub-model. Following [15], we choose $\mathcal{R}(\cdot)$ as: $\mathcal{R}(T(a), p^{(m)}T_{tol}) = \log(max(T(a), p^{(m)}T_{tol})/p^{(m)}T_{tol})$. In summary, we choose the 'best' sub-model which keeps the loss value small and meanwhile, satisfies the FLOPS constraint. Note that this is in contrast to the data-independent sub-model selection methods [11, 6, 1].

Next for the encode operator $Enc\{\cdot\}$, we follow the definition of Eq. (2) to have:

$$Enc^{(m)}\{\cdot\} = -\nabla_\omega \theta_\omega^{(m)} = -\nabla_\omega \big(a_\omega^{(m)} \odot \omega\big) = -Diag\{a_\omega^{(m)}\} - Diag\{\omega\}\nabla_\omega a_\omega^{(m)} \quad (8)$$

Where the last equality follows from the chain rule. Finally, combine Eq. (7) and Eq. (8), we can solve Eq. (6) with Algorithm 1.

*Remark* 3. For the encode operator Eq. (8), we have $Enc^{(m)}(\cdot) \approx -Diag\{a_\omega^{(m)}\}$ by omitting the second term in Eq. (8), where $\Delta\theta$ denotes the updates to the sub-model $\theta$. As shown in Line 21 of Algorithm 1, we have the update of the server model be $\Delta\omega = \frac{1}{|\mathcal{M}_t|} \sum_{m \in \mathcal{M}_t} \eta_g Diag\{a_\omega^{(m)}\}\Delta\theta$. In other words, the server simply aggregates the updates of the sub-model, this is the aggregation rule widely used in the partial training literature [11, 6, 1].

## 5 Numerical Experiments

In this section, we perform numerical experiments to validate the efficacy of FedSep in solving communication-efficient FL and model-heterogeneous FL as discussed in Section 4. The code is written with Pytorch [39], and the Federated Learning environment is simulated via Pytorch.Distributed Package. We used servers with AMD EPYC 7763 64-core CPU and 8 NVIDIA V100 GPUs to run our experiments.

### 5.1 Experiments for Communication-Efficient Federated Learning

In this section, we provide numerical experiments for the communication-efficient FL task discussed in Section 4.1. We first introduce the experimental settings.

**Dataset.** We consider MNIST [29] and CIFAR-10 [27] in our experiments. We create the federated version of these datasets by evenly distributing the training samples among 10 clients. We consider an I.I.D setting and a Non-I.I.D setting. For the I.I.D setting, each client has data samples from all 10 classes, while for the Non-I.I.D setting, each client only has data samples from 2 classes.

**Model.** For experiments related to MNIST, we use a four-layer convolutional neural network. The first three layers contain 32 kernels of size 3×3 and the fourth layer contains 32 kernels of size 2×2. For the experiments related to CIFAR-10, we increase the number of kernels to 64. The total number of parameters of the model is around $10^5$. For the sketch matrix $S^{(m)}$ of our FedSep, we let all clients choose the same sketch matrix, and we test the dimension of the communication parameter $p \in \{10, 100, 500, 1000, 2000, 5000, 10000, 20000\}$.

**Baselines and Hyper-parameters.** For baselines, we first compare with the uncompressed baseline (the communication parameter and the learning parameter are in the same dimension); Next we compare with the local Top-K compressor (with error feedback [44]) and the Count-Sketch compressor [42]. Please refer to the Appendix for the hyper-parameter selection of FedSep and baselines.

Table 1: **Test accuracy comparison between FedSep with other model-heterogeneous FL baseline methods.** High data heterogeneity represents $K = 2$ for CIFAR-10 and $K = 20$ for CIFAR-100; Lower data heterogeneity represents $K = 5$ for CIFAR-10 and $K = 50$ for CIFAR-100.

| | Method | High Data Heterogeneity | | Low Data Heterogeneity | |
|---|---|---|---|---|---|
| | | **CIFAR-10** | **CIFAR-100** | **CIFAR-10** | **CIFAR-100** |
| KD-based | FedDF [34] | 73.81 (± 0.42) | 31.87 (± 0.46) | 76.55 (± 0.32) | 37.87 (± 0.31) |
| | DS-FL [22] | 65.27 (± 0.53) | 29.12 (± 0.51) | 68.44 (± 0.47) | 33.56 (± 0.55) |
| | Fed-ET [9] | **78.66 (± 0.31)** | **35.78 (± 0.45)** | **81.13 (± 0.28)** | **41.58 (± 0.36)** |
| PT-based | HeteroFL [11] | 63.90 (± 2.74) | 52.38 (± 0.80) | 73.19 (± 1.71) | 57.44 (± 0.42) |
| | Federated Dropout [6] | 46.64 (± 3.05) | 45.07 (± 0.07) | 76.20 (± 2.53) | 46.40 (± 0.21) |
| | ZeroFL [41] | 64.61 (± 2.18) | 51.39 (± 0.45) | 83.31 (± 0.78) | 53.62 (± 0.51) |
| | FedDST [5] | 67.65 (± 1.27) | 54.21 (± 0.34) | 84.57 (± 0.28) | 54.97 (± 0.44) |
| | Flash [2] | 67.08 (± 1.46) | 54.92 (± 0.29) | 84.61 (± 0.37) | 55.04 (± 0.32) |
| | FedRolex [1] | 69.44 (± 1.50) | 56.57 (± 0.15) | 84.45 (± 0.36) | 58.73 (± 0.33) |
| | **FedSep** (Ours) | **71.13 (± 0.94)** | **58.16 (± 0.25)** | 84.61 (± 0.37) | **61.41 (± 0.29)** |
| | Homogeneous (smallest) | 38.82 (± 0.88) | 12.69 (± 0.50) | 46.86 (± 0.54) | 19.70 (± 0.34) |
| | Homogeneous (largest) | **75.74 (± 0.42)** | **60.89 (± 0.60)** | **84.48 (± 0.58)** | **62.51 (± 0.20)** |

The experimental results are summarized in Figure 2 for MNIST (Figure 3 in the Appendix for CIFAR-10). As shown by the figures, FedSep outperforms other baselines. In particular, our FedSep achieves much better performance in the high-compression-rate regime and can get similar performance as other baselines in the low-compression-rate regime. Please refer to the Appendix for more ablation studies.

## 5.2 Experiments for Model-Heterogeneous Federated Learning

In this section, we provide numerical experiments for the task of a model-heterogeneous FL task discussed in Section 4.2. First, we introduce the experimental settings.

**Dataset.** We consider CIFAR-10 and CIFAR-100 [27] in our experiments. We create Federated datasets by evenly distributing images among clients; we have 100 clients in our experiments. We create data heterogeneity by controlling the number of classes $K$ a client can have [1]. For CIFAR-10, we test $K \in \{2, 4, 5, 8\}$, while for CIFAR-100, we test $K \in \{5, 10, 20, 50\}$. Note that smaller values of $K$ mean higher heterogeneity of the data. For FedSep, the validation and training set are the same.

**Model.** We choose ResNet-18 [17] in the experiments. Following the setting in [1, 11], we replace the batch normalization layer with static batch normalization and add a scalar module after each convolution layer. Instead of using the coordinate mask as defined in Eq. (6), we select kernels in each convolutional layer of ResNet-18; furthermore, we parameterize the mask $a$ with a neural network $HN(\phi)$ [15], *i.e.* $a = HN(\phi)$; finally, we reuse the masks and only update the masks every two epochs, this stabilizes the training. This reduces the dimension of $a$ that we need to optimize. In experiments, the size of the submodel $p$ (Eq. (6)) of a client is randomly chosen from $\{1, 0.5, 0.25, 0.125, 0.0625\}$, where the ratio $p$ is *w.r.t* the server model. For our FedSep, after optimizing Eq. (7), we select the top $p$ percent kernels of each convolutional layer by the value of the leaned mask $a$, and then we only use the selected kernels to perform the training. In the encode operation, we only evaluate the first term of Eq. (8) as stated in Remark 3.

**Baselines and Hyper-parameters.** We compare with both state-of-the-art Knowledge Distillation-based methods: FedDF [34], DS-FL [22] and Fed-ET [9], and Partial Training Based methods: HeteroFL [11], ZeroFL [41], Federated Dropout [6], FedDST [5], Flash [2] and FedRolex [1]. Fjord [19] gets similar performance as HeteroFL, so we do not include it in the comparision. For ZeroFL, FedDST and Flash, we consider their heterogeneous-model version by varying the compression rate among clients.

The experimental results are summarized in Table 1. As shown in the table, our FedSep gets comparable performance with the Knowledge Distillation based methods, which needs additional public data, furthermore, FedSep outperforms the partial training based methods, including the state-of-the-art FedRolex [1] method. This result shows the effectiveness of the adaptive sub-model extraction strategy used by our FedSep. Our method chooses the most important part of the model

at each step for clients, thus our method converges faster than other data-independent methods. For more experimental results, please refer to the Appendix.

## 6 Conclusion

In this work, we propose a novel federated learning framework, *i.e.* FedSep with separated communication and learning components. Based on the observation that communication and learning set opposite requirements to the parameter, we let clients to have two layers: a communication layer and a learning layer. We formulate the decode operation from the communication layer to the learning layer as solving a minimization problem, therefore, FedSep has a bilevel structure. We propose an efficient algorithm to solve this bilevel problem and also prove that the algorithms converge to a stationary point with rate $O(\epsilon^{-2})$. Finally, we apply FedSep to solve the communication-efficient FL task and the model heterogeneous FL task. Numerical experiments show the superior performance of our FedSep over various baselines.

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
