In this section, we include more details related to the Communication-efficient FL. We first introduce details of some definitions.

**Soft-threshold Operator.** We use the definition of the soft threshold operator $Q_{\gamma\beta}(\cdot)$ in Section 4.1, formally, we have the following definition:

$$Q_{\gamma\lambda}(x) = sign(x)max(abs(x) - \gamma\lambda, 0) \tag{9}$$

**Proximal Gradient Descent.** we solve $\theta_\omega^{(m)}$ through the following update rule:

$$\theta^+ = Q_{\gamma\beta}\big(\theta + \gamma(S^{(m)})^T(\omega - S^{(m)}\theta)\big) \tag{10}$$

where $Q_{\gamma\beta}(\cdot)$ is the soft-threshold operator and its definition is in Eq. (9) and $\gamma$ is some constant. We denote the generalized gradient based on Eq. (10) as:

$$G^*(\theta) = \frac{1}{\gamma}\big(\theta - \theta^+\big) \tag{11}$$

In summary, for the decode operation $Dec\{\cdot\}$ in Eq. (4), we solve decode problem through gradient descent with the gradient operator defined by Eq. (11). As for the encode operation, we have $G^*(\theta_\omega^{(m)}) = 0$ by definition of $\theta_\omega^{(m)}$, then take derivation over $\omega$ on both sides, and use the chain rule, we get the expression in Eq. (5).

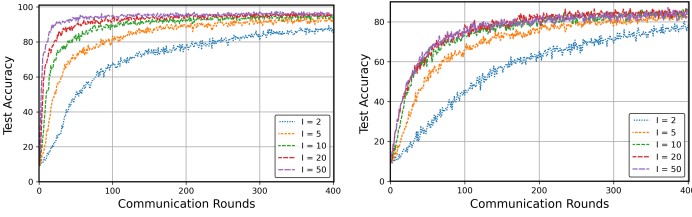

Figure 4: Test Accuracy w.r.t Communication Round for FedSep over the MNIST Dataset. The left figures shows results under the I.I.D case, and the right figure show results for the Non-I.I.D case. We vary the value of local step $I \in \{2, 5, 10, 20, 50\}$ in the figure, and we fix the dimension of the communication parameter as $p = 2000$.

Next, we introduce more details of experimental settings.

**Baselines and Hyper-Parameter.** We first compare with the uncompressed baseline (the communication parameter and the learning parameter are in the same dimension); Next we compare with the local Top-K compressor and the Count-Sketch compressor [48]. For the count-sketch, we tune the size of sketch table; a typical good value for the length of the sketch table is half of the communication parameter dimension. For our FedSep, we tune the values of $\beta$ and $\gamma$. A typical good value for

$\gamma$ is $[0.5, 5]$ and it depends on the dimension of the communication parameter $p$, for $\beta$, we choose $[0, 0.01]$. We use the SGD optimizer to optimize $\mathcal{L}(\theta_\omega^{(m)}; \mathcal{D}_{tr}^{(m)})$ for all methods, and set the learning rate at $0.1$, and set $I_{dec} = 10$ for the decode operation, *i.e.* for solving the Lasso problem.

**More Experimental Results.** The experimental results for the CIFAR-10 dataset is included in Figure 3. We observe that FedSep also outperforms other baselines as in the MNIST data set. Finally, in Figure 4, we vary the number of local steps $I$ under a fixed compression rate. As shown in the figure, our FedSep can benefit from more local steps. More specifically, for the homogeneous case, our FedSep can benefit from more local steps as large as $I = 50$, while for the heterogeneous case, increasing local steps brings little benefit to $I > 10$.

## A.2 Model-Heterogeneous FL

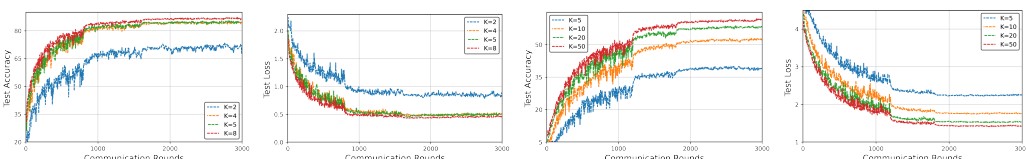

Figure 5: Test Accuracy and Loss w.r.t Communication Rounds for FedSep under different levels of data heterogeneity. $K$ is the number of classes each client has. The first two plots show results for CIFAR-10, then the last two plots show results for CIFAR-100.

In this section, we introduce more details of the experimental setting for Model-Heterogeneous FL task and present more experimentsl results.

**Hyper-parameters.** For our FedSep, we perform grid search to search the best parameter setting. More specifically, we decode problem Eq. (7) with $\beta = 10$ and AdamW [40] with learning rate 0.01. Regarding the learning problem, for CIFAR-10, we use the Adam optimizer with learning rate $10^{-4}$ when $K = \{2, 4\}$ and $2 \times 10^{-4}$ when $K \in \{5, 8\}$, for CIFAR-100, we use the Adam optimizer with learning rate $2 \times 10^{-4}$ when $K = \{5, 10, 20, 50\}$. We decrease learning by a factor of 0.25 at the $\{800, 1600\}$ steps for the CIFAR-10 dataset and at the $\{1200, 1800\}$ global steps for the CIFAR-100 dataset.

**More Experimental Results.** Next, we test our FedSep at different levels of heterogeneity. The results are summarized in Figure 5. As shown by the figures, our FedSep is quite robust when clients have highly heterogeneous data.

# B  More Related Works

**Federated Learning.** FL is a promising paradigm for performing machine learning tasks on distributed located data. Compared to traditional distributed learning in the data center, FL poses new challenges such as heterogeneity [27, 50, 43, 37], privacy [44, 42, 57] and communication bottleneck [58, 39, 54, 28, 25]. In particular, communication cost is one of the major bottlenecks in FL. A widely used approach to reduce communication burden is through compression: clients compress parameters before transmitting to the server. Various compressors [58, 39, 54, 28] have been studied in the literature. In particular, the local Top-K compressor selects the $K$ largest coordinates of the parameter to transmit. From the view of FedSep: the vector of Top-K coordinates is the communication parameter, while the encode operation is finding these coordinates. Count Sketch [7] based compressors [25, 48] are also widely studied. Since the count sketch is linear, from the point of view of FedSep: count sketch performs a linear mapping between the communication parameter and the learning parameter. Next, model-heterogeneous FL is also studied to solve the device heterogeneity issue. There are two main categories of heterogeneous model methods: knowledge-distillation (KD) [38, 24, 10] based and partial-training (PT) [12, 6, 1] based. KD-based methods treat the server model as the teacher and the clients' model as the student. A disadvantage of this category is the dependence on a public dataset. In contrast, PT-based methods select a subset of the server model for each client, and in the aggregation phase, the server simply aggregates the updated sub-models. We also develop a PT-based method where we select the sub-models adaptively based on the clients' local data.

**Bilevel Optimization.** Bilevel problem is a type of two-layer optimization problem. It includes an upper-level problem and a lower-level problem, and the upper-level problem relies on the solution of the lower-level problems. The study of bilevel problems dates back at least to the 1960s [59] and is followed by [15, 52, 61, 49]. In the machine learning community, Hyper-parameter Optimization problems [31, 8, 4, 13] are one of the most studied bilevel problems. Early bilevel optimization algorithms solved the lower problem exactly to update the upper variable. Recently, researchers developed algorithms that solve the lower problem approximately with a fixed number of steps and use the back-propagation technique to compute the approximate hyper-gradient [14, 41, 16, 46, 51]. More recently, single-loop algorithms [18, 20, 26, 29, 9, 62, 34, 11, 22, 23] based on alternative updates of lower and upper level problems are proposed. However, less work studies bilevel problems under the federated setting. [55, 36, 35] studies a type of federated bilevel problems where the lower level problems are federated. In particular, [36] solves the noisy label problem in FL. Note that our FedSep is formulated as a type of federated bilevel problems, however, our problem is different from that in [55, 36], as our problem has a unique lower problem for each client.

## C   Proof for the convergence of Algorithm 1

In this section, we present the proofs for Algorithm 1. First, we denote some notation for clarity. Recall that we denote $\Delta \hat{y}^{(m)} = \hat{y}_I^{(m)} - \hat{y}_0^{(m)} = \sum_{i=0}^{I-1} -\eta \nabla f^{(m)}(\hat{y}_i^{(m)}; \mathcal{B}_{\hat{y}})$ to be the update of the learning problem on client $m$. Then we denote $\mu_i^{(m)}$ as:

$$\mu_i^{(m)} = -\nabla_{xy} g^{(m)}(x, y_{I_{dec}}^{(m)}) \big(\nabla_{yy} g^{(m)}(x, y_{I_{dec}}^{(m)})\big)^{-1} \nabla f^{(m)}(\hat{y}_i^{(m)})$$

and denote $\mu^{(m)}$ as:

$$\mu^{(m)} = -\nabla_{xy} g^{(m)}(x, y_{I_{dec}}^{(m)}) \big(\nabla_{yy} g^{(m)}(x, y_{I_{dec}}^{(m)})\big)^{-1} \Delta \hat{y}^{(m)}$$

while in Algorithm 1, we use the stochastic encoder $\widetilde{Enc}\{\cdot\}$ to compute $\Delta \hat{x}_t^{(m)}$,

$$\Delta \hat{x}_t^{(m)} = \sum_{q=0}^{Q} \tau \nabla_{xy} g^{(m)}(x, y_{I_{dec}}^{(m)}; \mathcal{B}_x)(I - \tau \nabla_{yy} g^{(m)}(x, y_{I_{dec}}^{(m)}; \mathcal{B}_x))^q \Delta \hat{y}^{(m)}$$

Since $x^{(m)}$ is not updated during local steps, we omit the superscript when clear from the context. We further denote $\Delta \hat{x}_{t,i}^{(m)}$ as

$$\Delta \hat{x}_{t,i}^{(m)} = -\sum_{q=0}^{Q} \tau \nabla_{xy} g^{(m)}(x, y_{I_{dec}}^{(m)}; \mathcal{B}_x)(I - \tau \nabla_{yy} g^{(m)}(x, y_{I_{dec}}^{(m)}; \mathcal{B}_x))^q \nabla f^{(m)}(\hat{y}_i^{(m)}; \mathcal{B}_{\hat{y}})$$

Lastly, recall the hyper-gradient of $h^{(m)}(x)$ has the form of:

$$\nabla h^{(m)}(x) = -\nabla_{xy} g^{(m)}(x, y_x^{(m)}) \big(\nabla_{yy} g^{(m)}(x, y_x^{(m)})\big)^{-1} \nabla f^{(m)}(y_x^{(m)})$$

**Assumption C.1** (Assumption 4). We have unbiased stochastic first order and second order derivative oracle with bounded variance, more specifically, denote $z = (x, y)$, we have:

  a) we have $\nabla f^{(m)}(y; \xi)$, such that: $E[\nabla f^{(m)}(y; \xi)] = \nabla f^{(m)}(y)$ and $var(\nabla f^{(m)}(y; \xi)) \leq \sigma^2$;

  b) we have $\nabla g^{(m)}(z; \xi)$, such that: $E[\nabla g^{(m)}(z; \xi)] = \nabla g^{(m)}(z)$ and $var(\nabla g^{(m)}(z; \xi)) \leq \sigma^2$;

  c) we have $\nabla_{y^2} g^{(m)}(z, \xi)$, such that: $E[\nabla_{y^2} g^{(m)}(z; \xi)] = \nabla_{y^2} g^{(m)}(z)$ and $var(\nabla_{y^2} g^{(m)}(z; \xi)) \leq \sigma^2$;

  d) we have $\nabla_{xy} g^{(m)}(z; \xi)$, such that: $E[\nabla_{xy} g^{(m)}(z; \xi)] = \nabla_{xy} g^{(m)}(z)$ and $var(\nabla_{xy} g^{(m)}(z; \xi)) \leq \sigma^2$;

**Proposition C.2.** *Suppose Assumptions 3.2 and 3.3 hold, the following statements hold:*

  a) $\|\mu_i^{(m)} - \nabla h^{(m)}(x)\|^2 \leq 2\hat{L}^2 \|y_{I_{dec}}^{(m)} - y_x^{(m)}\|^2 + 2\kappa^2 L^2 \|\hat{y}_i^{(m)} - y_x^{(m)}\|^2$, *where* $\hat{L} = O(\kappa^2)$.

b) $h^{(m)}(x)$ is Lipschitz continuous in $x$ with constant $\bar{L}$ i.e., for any given $x_1, x_2 \in \mathbb{R}^p$, we have $\|\nabla h^{(m)}(x_2) - \nabla h^{(m)}(x_1)\|^2 \le \bar{L}^2 \|x_2 - x_1\|$ where $\bar{L} = O(\kappa^3)$.

where we denote the condition number as $\kappa = L/\mu$.

*Proof.* We prove the Part a) here. Proof of b) and can be referred in Lemma 2.2 of [18].

$$\|\mu_i^{(m)} - \nabla h^{(m)}(x)\|$$

$$= \left\| \nabla_{xy} g(x, y_{I_{dec}}^{(m)}) \left( \nabla_{yy} g(x, y_{I_{dec}}^{(m)}) \right)^{-1} \nabla f^{(m)}(\hat{y}_i^{(m)}) - \nabla_{xy} g(x, y_x^{(m)}) \left( \nabla_{yy} g(x, y_x^{(m)}) \right)^{-1} \nabla f^{(m)}(y_x^{(m)}) \right\|$$

$$\le \left\| \nabla_{xy} g(x, y_{I_{dec}}^{(m)}) - \nabla_{xy} g(x, y_x^{(m)}) \right\| \left\| \left( \nabla_{yy} g(x, y_x^{(m)}) \right)^{-1} \nabla f^{(m)}(\hat{y}_i^{(m)}) \right\|$$

$$+ \left\| \nabla_{xy} g(x, y_x^{(m)}) \right\| \left\| \left( \nabla_{yy} g(x, y_{I_{dec}}^{(m)}) \right)^{-1} \nabla f^{(m)}(\hat{y}_i^{(m)}) - \left( \nabla_{yy} g(x, y_x^{(m)}) \right)^{-1} \nabla f^{(m)}(y_x^{(m)}) \right\|$$

$$\le \frac{C_f L_{xy}}{\mu} \left\| y_{I_{dec}}^{(m)} - y_x^{(m)} \right\| + L \left\| \left( \nabla_{yy} g(x, y_{I_{dec}}^{(m)}) \right)^{-1} \nabla f^{(m)}(\hat{y}_i^{(m)}) - \left( \nabla_{yy} g(x, y_x^{(m)}) \right)^{-1} \nabla f^{(m)}(y_x^{(m)}) \right\|$$

$$\le \frac{C_f L_{xy}}{\mu} \left\| y_{I_{dec}}^{(m)} - y_x^{(m)} \right\| + C_f L \left\| \left( \nabla_{yy} g(x, y_{I_{dec}}^{(m)}) \right)^{-1} - \left( \nabla_{yy} g(x, y_x^{(m)}) \right)^{-1} \right\| + \kappa \left\| \nabla f^{(m)}(\hat{y}_i^{(m)}) - \nabla f^{(m)}(y_x^{(m)}) \right\|$$

$$\le \left( \frac{C_f L_{xy}}{\mu} + \frac{\kappa C_f L_{yy}}{\mu} \right) \left\| y_{I_{dec}}^{(m)} - y_x^{(m)} \right\| + \kappa \left\| \nabla f^{(m)}(\hat{y}_i^{(m)}) - \nabla f^{(m)}(y_x^{(m)}) \right\|$$

Suppose we denote $\hat{L} = \left( \frac{C_f L_{xy}}{\mu} + \frac{\kappa C_f L_{yy}}{\mu} \right)$, then we have:

$$\left\| \mu_i^{(m)} - \nabla h^{(m)}(x) \right\|^2 \le 2\hat{L}^2 \left\| y_{I_{dec}}^{(m)} - y_x^{(m)} \right\|^2 + 2\kappa^2 \left\| \Delta \nabla f^{(m)}(\hat{y}_i^{(m)}) - \nabla_y f(y_x^{(m)}) \right\|^2$$

which completes the proof. $\square$

**Proposition C.3.** *(Lemma 4 and 7 in [62]) Suppose Assumptions 3.2, 3.3 and 3.4 hold and $\tau < \frac{1}{L}$, we have*

a) $\|\mathbb{E}_\xi [\Delta \hat{x}_{t,i}^{(m)}] - \mu_i^{(m)}\| \le G_1$, where $G_1 = \kappa(1 - \tau\mu)^{Q+1} C_f$

b) $\mathbb{E}\|\Delta \hat{x}_{t,i}^{(m)} - \mathbb{E}_\xi [\Delta \hat{x}_{t,i}^{(m)}]\|^2 \le G_2^2$, where $G_2^2 = (2C_f^2 + 12C_f^2 L^2 \tau^2 (Q+1)^2 + 4C_f^2 L^2 (Q+2)(Q+1)^2 \tau^4 \sigma^2)/b_x$

where we assume the mini-batch has $|\mathcal{B}_x| = b_x$

## C.1 Lower Problem Solution Error

**Lemma C.4.** *When $\gamma < \frac{1}{L}$, we have:*

$$\frac{1}{M} \sum_{m=1}^{M} \mathbb{E} \left\| y_{I_{dec}}^{(m)} - y_{x^{(m)}}^{(m)} \right\|^2 \le \frac{(1 - \mu\gamma)^{I_{dec}}}{M} \sum_{m=1}^{M} \mathbb{E} \left\| y_0^{(m)} - y_{x^{(m)}}^{(m)} \right\|^2 + \frac{2\gamma\sigma^2}{\mu b_y}$$

*Proof.* First, follow the property of the strong convex function, we have:

$$\mathbb{E} \left\| y_i^{(m)} - y_{x^{(m)}}^{(m)} \right\|^2 \le (1 - \mu\gamma) \mathbb{E} \left\| y_{i-1}^{(m)} - y_{x^{(m)}}^{(m)} \right\|^2 + \frac{2\gamma^2 \sigma^2}{b_y}$$

where we choose $\gamma < 1/L$. Next, we telescope from $i = 1 \to I_{dec}$ to have

$$\mathbb{E} \left\| y_{I_{dec}}^{(m)} - y_{x^{(m)}}^{(m)} \right\|^2 \le (1 - \mu\gamma)^{I_{dec}} \mathbb{E} \left\| y_0^{(m)} - y_{x^{(m)}}^{(m)} \right\|^2 + \frac{2\gamma^2 \sigma^2}{b_y} \sum_{i=0}^{I} (1 - \mu\gamma)^i$$

$$\le (1 - \mu\gamma)^{I_{dec}} \mathbb{E} \left\| y_0^{(m)} - y_{x^{(m)}}^{(m)} \right\|^2 + \frac{2\gamma\sigma^2}{\mu b_y}$$

Average over all clients, we get the claim in the lemma. $\square$

**Lemma C.5.** *For $I \geq 1$, than we have:*

$$\frac{1}{M} \sum_{m=1}^{M} \sum_{i=1}^{I} \mathbb{E}\|\hat{y}_i^{(m)} - y_x^{(m)}\|^2 \leq \frac{3I}{M} \sum_{m=1}^{M} \mathbb{E}\|\hat{y}_0^{(m)} - y_x^{(m)}\|^2 + 6I^2\eta^2 \frac{1}{M} \sum_{m=1}^{M} \sum_{i=1}^{I} \|\nabla f^{(m)}(\hat{y}_i^{(m)})\|^2 + \frac{2I^3\eta^2\sigma^2}{b_y}$$

*Proof.* By the update rule in Algorithm 1, we have:

$$\mathbb{E}\|\hat{y}_i^{(m)} - y_x^{(m)}\|^2 = \mathbb{E}\|\hat{y}_{i-1}^{(m)} - \eta\nabla f^{(m)}(\hat{y}_{i-1}^{(m)}; \mathcal{B}_{\hat{y}}) - y_x^{(m)}\|^2$$

$$\leq (1 + \frac{1}{I})\mathbb{E}\|\hat{y}_{i-1}^{(m)} - y_x^{(m)}\|^2 + (1 + I)\eta^2\|\nabla f^{(m)}(\hat{y}_{i-1}^{(m)}; \mathcal{B}_{\hat{y}})\|^2$$

$$\leq (1 + \frac{1}{I})\mathbb{E}\|\hat{y}_{i-1}^{(m)} - y_x^{(m)}\|^2 + 2I\eta^2\|\nabla f^{(m)}(\hat{y}_{i-1}^{(m)}; \mathcal{B}_{\hat{y}})\|^2$$

$$\leq (1 + \frac{1}{I})\mathbb{E}\|\hat{y}_{i-1}^{(m)} - y_x^{(m)}\|^2 + 2I\eta^2\|\nabla f^{(m)}(\hat{y}_{i-1}^{(m)})\|^2 + \frac{2I\eta^2\sigma^2}{b_y}$$

where the first inequality is by the generalized inequality. Next we telescope over $i$, to obtain:

$$\mathbb{E}\|\hat{y}_i^{(m)} - y_x^{(m)}\|^2 \leq \sum_{j=1}^{i}(1 + \frac{1}{I})^{i-j}\left(2I\eta^2\|\nabla f^{(m)}(\hat{y}_j^{(m)})\|^2 + \frac{2I\eta^2\sigma^2}{b_y}\right) + (1 + \frac{1}{I})^i\mathbb{E}\|\hat{y}_0^{(m)} - y_x^{(m)}\|^2$$

$$\leq (1 + \frac{1}{I})^I \sum_{i=1}^{I}\left(2I\eta^2\|\nabla f^{(m)}(\hat{y}_i^{(m)})\|^2 + \frac{2I\eta^2\sigma^2}{b_y}\right) + (1 + \frac{1}{I})^I\mathbb{E}\|\hat{y}_0^{(m)} - y_x^{(m)}\|^2$$

$$\leq 3\mathbb{E}\|\hat{y}_0^{(m)} - y_x^{(m)}\|^2 + 6I\eta^2 \sum_{i=1}^{I}\|\nabla f^{(m)}(\hat{y}_i^{(m)})\|^2 + \frac{2I^2\eta^2\sigma^2}{b_y}$$

The third inequality uses the inequality $log(1+a/x) \leq a/x$ for $x > -a$, so we have $(1+a/x)^x \leq e^a$, Then we choose $a = 1$ and $x = I$. Finally, we use the fact that $e \leq 3$. It completes the proof by summing over all $i$. $\square$

### C.2   Descent Lemma

**Lemma C.6.** *For all $t \in [T]$, the iterates generated satisfy:*

$$\mathbb{E}\left\|\nabla h(x_t) - \mathbb{E}_\xi[\bar{\Delta}\hat{x}_{t,i}]\right\|^2 \leq \frac{1}{M} \sum_{m=1}^{M} \left(2\hat{L}^2\mathbb{E}\|y_{I_{dec}}^{(m)} - y_x^{(m)}\|^2 + 2\kappa^2 L^2 \mathbb{E}\|\hat{y}_i^{(m)} - y_x^{(m)}\|^2\right) + 2G_1^2$$

*Proof.* By definition of $\bar{\Delta}\hat{x}_{t,i}$, $\mu_{t,i}^{(m)}$ and $\nabla h(x_t)$, we have:

$$\mathbb{E}\left\|\nabla h(x_t) - \mathbb{E}_\xi[\bar{\Delta}\hat{x}_{t,i}]\right\|^2$$

$$\overset{(a)}{\leq} \frac{1}{M} \sum_{m=1}^{M} \mathbb{E}\left\|\mathbb{E}_\xi[\Delta\hat{x}_t^{(m)}] - \nabla h^{(m)}(x_t)\right\|^2$$

$$\leq \frac{2}{M} \sum_{m=1}^{M} \mathbb{E}\left[\left\|\mathbb{E}_\xi[\Delta\hat{x}_{t,i}^{(m)}] - \mu_{t,i}^{(m)}\right\|^2 + \left\|\mu_{t,i}^{(m)} - \nabla h^{(m)}(x_t)\right\|^2\right]$$

$$\overset{(b)}{\leq} \frac{1}{M} \sum_{m=1}^{M} \left(2\hat{L}^2\mathbb{E}\|y_{I_{dec}}^{(m)} - y_x^{(m)}\|^2 + 2\kappa^2\mathbb{E}\|\nabla f^{(m)}(\hat{y}_i^{(m)}) - \nabla f^{(m)}(y_x^{(m)})\|^2\right) + 2G_1^2$$

$$\leq \frac{1}{M} \sum_{m=1}^{M} \left(2\hat{L}^2\mathbb{E}\|y_{I_{dec}}^{(m)} - y_x^{(m)}\|^2 + 2\kappa^2 L^2\mathbb{E}\|\hat{y}_i^{(m)} - y_x^{(m)}\|^2\right) + 2G_1^2$$

where inequality (a) follows the generalized triangle inequality; inequality (b) follows the Proposition C.2 and Proposition C.3.

$\square$

**Lemma C.7.** *Suppose $\eta\eta_g \leq \frac{1}{2I\bar{L}}$ For $t \in [T]$, the iterates generated satisfy:*

$$\mathbb{E}[h(x_{t+1})] \leq \mathbb{E}[h(x_t)] - \frac{I\eta\eta_g}{2}\mathbb{E}\|\nabla h(x_t)\|^2 - \frac{\eta\eta_g}{4}\sum_{i=1}^{I}\mathbb{E}\left\|\mathbb{E}_\xi[\bar{\Delta}\hat{x}_{t,i}]\right\|^2 + \frac{I\eta^2\eta_g^2\bar{L}G_2^2}{2b_xM} + I\eta\eta_g G_1^2$$

$$+ \frac{\eta\eta_g}{M}\sum_{m=1}^{M}\sum_{i=1}^{I}\left(\hat{L}^2\mathbb{E}\|y_{I_{dec}}^{(m)} - y_x^{(m)}\|^2 + \kappa^2L^2\mathbb{E}\|\hat{y}_i^{(m)} - y_x^{(m)}\|^2\right)$$

*where the expectation is w.r.t the stochasticity of the algorithm.*

*Proof.* Using the smoothness of $f$ we have:

$$\mathbb{E}[h(x_{t+1})] \leq \mathbb{E}[h(x_t)] + \mathbb{E}\langle\nabla h(x_t), x_{t+1} - x_t\rangle + \frac{\bar{L}}{2}\mathbb{E}\|x_{t+1} - x_t\|^2$$

$$\overset{(a)}{=} \mathbb{E}[h(x_t)] - \eta_g\mathbb{E}\langle\nabla h(x_t), \mathbb{E}_\xi[\bar{\Delta}\hat{x}_t]\rangle + \frac{\eta_g^2\bar{L}}{2}\mathbb{E}\|\mathbb{E}_\xi[\bar{\Delta}\hat{x}_t]\|^2 + \frac{I\eta^2\eta_g^2\bar{L}G_2^2}{2b_xM}$$

$$= \mathbb{E}[h(x_t)] - \eta_g\eta\sum_{i=1}^{I}\mathbb{E}\langle\nabla h(x_t), \mathbb{E}_\xi[\bar{\Delta}\hat{x}_{t,i}]\rangle + \frac{I\eta^2\eta_g^2\bar{L}}{2}\sum_{i=1}^{I}\mathbb{E}\|\mathbb{E}_\xi[\bar{\Delta}\hat{x}_{t,i}]\|^2 + \frac{I\eta^2\eta_g^2\bar{L}G_2^2}{2b_xM}$$

$$\overset{(b)}{=} \mathbb{E}[h(x_t)] - \frac{I\eta\eta_g}{2}\mathbb{E}\|\nabla h(x_t)\|^2 + \frac{\eta\eta_g}{2}\sum_{i=1}^{I}\mathbb{E}\|\nabla h(x_t) - \mathbb{E}_\xi[\bar{\Delta}\hat{x}_{t,i}]\|^2$$

$$- \left(\frac{\eta\eta_g}{2} - \frac{I\eta^2\eta_g^2\bar{L}}{2}\right)\sum_{i=1}^{I}\mathbb{E}\|\mathbb{E}_\xi[\bar{\Delta}\hat{x}_{t,i}]\|^2 + \frac{I\eta^2\eta_g^2\bar{L}G_2^2}{2b_xM}$$

$$\overset{(c)}{\leq} \mathbb{E}[h(x_t)] - \frac{I\eta\eta_g}{2}\mathbb{E}\|\nabla h(x_t)\|^2 - \frac{\eta\eta_g}{4}\sum_{i=1}^{I}\mathbb{E}\left\|\mathbb{E}_\xi[\bar{\Delta}\hat{x}_{t,i}]\right\|^2 + \frac{I\eta^2\eta_g^2\bar{L}G_2^2}{2b_xM} + I\eta\eta_g G_1^2$$

$$+ \frac{\eta\eta_g}{M}\sum_{m=1}^{M}\sum_{i=1}^{I}\left(\hat{L}^2\mathbb{E}\|y_{I_{dec}}^{(m)} - y_x^{(m)}\|^2 + \kappa^2L^2\mathbb{E}\|\hat{y}_i^{(m)} - y_x^{(m)}\|^2\right)$$

where equality $(a)$ follows from the iterate update given in Step 21 of Algorithm 1; $(b)$ uses $\langle a, b\rangle = \frac{1}{2}[\|a\|^2 + \|b\|^2 - \|a - b\|^2]$; (c) follows the condition that $I\eta\eta_g \leq 1/2\bar{L}$ and Lemma C.6. This completes the proof. $\qquad\square$

## C.3 Proof of Convergence Theorem

We prove the convergence of Algorithm 1 in this section.

**Theorem C.8.** *Suppose we the learning rates*

$$\eta = \min\left(1, \left(\frac{8Ib_xM\bar{L}h(x_1)}{TG_2^2}\right)^{1/2}, \left(\frac{4\bar{L}h(x_1)}{C_\eta I^2T}\right)^{1/3}\right), \ \gamma = \min\left(\frac{1}{2L}, \left(\frac{1}{C_\gamma T}\right)^{1/2}\right)$$

*and $\eta_g = \frac{1}{2I\bar{L}}$, then we have:*

$$\frac{1}{T}\sum_{t=1}^{T}\left(\mathbb{E}\|\nabla h(x_t)\|^2 + \frac{1}{2I}\sum_{i=1}^{I}\mathbb{E}\|\mathbb{E}_\xi[\bar{\Delta}\hat{x}_{t,i}]\|^2\right)$$

$$= O\left(\frac{\kappa^3}{T} + \left(\frac{\kappa^5}{T}\right)^{1/2} + \left(\frac{\kappa^6}{T^2}\right)^{1/3} + \kappa^2(1-\tau\mu)^{2(Q+1)} + \kappa^4(1-\mu\gamma)^{I_{dec}}\right)$$

*To reach an $\epsilon$ stationary point, we choose $Q = O(\kappa\log(\frac{\kappa}{\epsilon}))$, $I_{dec} = O(\kappa\log(\frac{\kappa}{\epsilon}))$ and $T = O(\kappa^3\epsilon^{-1.5})$ number of iterations.*

*Proof.* By Lemma C.7, and combine with Lemma C.4 and Lemma C.5 to have:

$$\mathbb{E}[h(x_{t+1})] \leq \mathbb{E}[h(x_t)] - \frac{I\eta\eta_g}{2}\mathbb{E}\|\nabla h(x_t)\|^2 - \frac{\eta\eta_g}{4}\sum_{i=1}^{I}\mathbb{E}\left\|\mathbb{E}_\xi[\bar{\Delta}\hat{x}_{t,i}]\right\|^2 + \frac{I\eta^2\eta_g^2\bar{L}G_2^2}{2b_x M} + I\eta\eta_g G_1^2$$

$$+ \frac{\eta\eta_g}{M}\sum_{m=1}^{M}\sum_{i=1}^{I}\left(\hat{L}^2\mathbb{E}\|y_{I_{dec}}^{(m)} - y_x^{(m)}\|^2 + \kappa^2 L^2\mathbb{E}\|\hat{y}_i^{(m)} - y_x^{(m)}\|^2\right)$$

$$\leq \mathbb{E}[h(x_t)] - \frac{I\eta\eta_g}{2}\mathbb{E}\|\nabla h(x_t)\|^2 - \frac{\eta\eta_g}{4}\sum_{i=1}^{I}\mathbb{E}\left\|\mathbb{E}_\xi[\bar{\Delta}\hat{x}_{t,i}]\right\|^2 + \frac{I\eta^2\eta_g^2\bar{L}G_2^2}{2b_x M} + I\eta\eta_g G_1^2$$

$$+ \frac{I\hat{L}^2\eta\eta_g(1-\mu\gamma)^{I_{dec}}}{M}\sum_{m=1}^{M}\mathbb{E}\left\|y_0^{(m)} - y_{x^{(m)}}^{(m)}\right\|^2 + \frac{2I\hat{L}^2\eta\eta_g\gamma\sigma^2}{\mu b_y} + \frac{2I^3\kappa^2 L^2\eta^3\eta_g\sigma^2}{b_y}$$

$$+ \frac{3I\kappa^2 L^2\eta\eta_g}{M}\sum_{m=1}^{M}\mathbb{E}\|\hat{y}_0^{(m)} - y_x^{(m)}\|^2 + \frac{6I^2\kappa^2 L^2\eta^3\eta_g}{M}\sum_{m=1}^{M}\sum_{i=1}^{I}\|\nabla f^{(m)}(\hat{y}_i^{(m)})\|^2$$

By Algorithm 1, we have $\hat{y}_0^{(m)} = y_{I_{dec}}^{(m)}$, then we have:

$$\mathbb{E}[h(x_{t+1})] \leq \mathbb{E}[h(x_t)] - \frac{I\eta\eta_g}{2}\mathbb{E}\|\nabla h(x_t)\|^2 - \frac{\eta\eta_g}{4}\sum_{i=1}^{I}\mathbb{E}\left\|\mathbb{E}_\xi[\bar{\Delta}\hat{x}_{t,i}]\right\|^2 + \frac{I\eta^2\eta_g^2\bar{L}G_2^2}{2b_x M} + I\eta\eta_g G_1^2$$

$$+ \frac{I(3\kappa^2 L^2 + \hat{L}^2)\eta\eta_g(1-\mu\gamma)^{I_{dec}}}{M}\sum_{m=1}^{M}\mathbb{E}\left\|y_0^{(m)} - y_{x^{(m)}}^{(m)}\right\|^2 + \frac{2I\hat{L}^2\eta\eta_g\gamma\sigma^2}{\mu b_y}$$

$$+ \frac{6I\kappa^2 L^2\eta\eta_g\gamma\sigma^2}{\mu b_y} + \frac{6I^2\kappa^2 L^2\eta^3\eta_g}{M}\sum_{m=1}^{M}\sum_{i=1}^{I}\|\nabla f^{(m)}(\hat{y}_i^{(m)})\|^2 + \frac{2I^3\kappa^2 L^2\eta^3\eta_g\sigma^2}{b_y}$$

Assume that $\|y_0^{(m)} - y_{x^{(m)}}^{(m)}\|^2 \leq C_0$ for some constant $C_0$, and use Assumption 3.2. We sum over $t \in [T]$ to obtain:

$$\frac{1}{T}\sum_{t=1}^{T}\left(\frac{I\eta\eta_g}{2}\mathbb{E}\|\nabla h(x_t)\|^2 + \frac{\eta\eta_g}{4}\sum_{i=1}^{I}\mathbb{E}\|\mathbb{E}_\xi[\bar{\Delta}\hat{x}_{t,i}]\|^2\right)$$

$$\leq \frac{h(x_1)}{T} + \frac{I\eta^2\eta_g^2\bar{L}G_2^2}{2b_x M} + I\eta\eta_g G_1^2 + I(3\kappa^2 L^2 + \hat{L}^2)\eta\eta_g(1-\mu\gamma)^{I_{dec}}C_0 + \frac{2I\hat{L}^2\eta\eta_g\gamma\sigma^2}{\mu b_y}$$

$$+ \frac{6I\kappa^2 L^2\eta\eta_g\gamma\sigma^2}{\mu b_y} + 6I^3\kappa^2 L^2\eta^3\eta_g C_f + \frac{2I^3\kappa^2 L^2\eta^3\eta_g\sigma^2}{b_y}$$

Then we divide by $(I\eta\eta_g)/2$ on both sides and have:

$$\frac{1}{T}\sum_{t=1}^{T}\left(\mathbb{E}\|\nabla h(x_t)\|^2 + \frac{1}{2I}\sum_{i=1}^{I}\mathbb{E}\|\mathbb{E}_\xi[\bar{\Delta}\hat{x}_{t,i}]\|^2\right)$$

$$\leq \frac{2h(x_1)}{TI\eta\eta_g} + \frac{\eta\eta_g\bar{L}G_2^2}{b_x M} + 12I^2\kappa^2 L^2\eta^2 C_f + \frac{4I^2\kappa^2 L^2\eta^2\sigma^2}{b_y}$$

$$+ \frac{4(3\kappa^2 L^2 + \hat{L}^2)\gamma\sigma^2}{\mu b_y} + 2G_1^2 + 2(3\kappa^2 L^2 + \hat{L}^2)(1-\mu\gamma)^{I_{dec}}C_0$$

Next, we denote constant $C_\eta = \left(12\kappa^2 L^2 C_f + \frac{4\kappa^2 L^2\sigma^2}{b_y}\right)$ and $C_\gamma = \frac{4(3\kappa^2 L^2 + \hat{L}^2)\sigma^2}{\mu b_y}$, and set $\eta_g = \frac{1}{2I\bar{L}}$, then we have:

$$\frac{1}{T}\sum_{t=1}^{T}\left(\mathbb{E}\|\nabla h(x_t)\|^2 + \frac{1}{2I}\sum_{i=1}^{I}\mathbb{E}\|\mathbb{E}_\xi[\bar{\Delta}\hat{x}_{t,i}]\|^2\right)$$

$$\leq \frac{4\bar{L}h(x_1)}{T\eta} + \frac{\eta G_2^2}{2I b_x M} + C_\eta I^2\eta^2 + C_\gamma\gamma + 2G_1^2 + 2(3\kappa^2 L^2 + \hat{L}^2)(1-\mu\gamma)^{I_{dec}}C_0$$

Then we choose

$$\eta = \min\left(1, \left(\frac{8Ib_xM\bar{L}h(x_1)}{TG_2^2}\right)^{1/2}, \left(\frac{4\bar{L}h(x_1)}{C_\eta I^2T}\right)^{1/3}\right), \ \gamma = \min\left(\frac{1}{2L}, \left(\frac{1}{C_\gamma T}\right)^{1/2}\right)$$

Then we obtain:

$$\frac{1}{T}\sum_{t=1}^{T}\left(\mathbb{E}\|\nabla h(x_t)\|^2 + \frac{1}{2I}\sum_{i=1}^{I}\mathbb{E}\|\mathbb{E}_\xi[\bar{\Delta}\hat{x}_{t,i}]\|^2\right)$$

$$\leq \frac{4\bar{L}h(x_1)}{T} + \left(\frac{2G_2^2\bar{L}h(x_1)}{Ib_xMT}\right)^{1/2} + \left(\frac{16I^2C_\eta\bar{L}^2h(x_1)^2}{T^2}\right)^{1/3}$$

$$+ \left(\frac{C_\gamma}{T}\right)^{1/2} + 2G_1^2 + 2(3\kappa^2L^2 + \hat{L}^2)(1-\mu\gamma)^{I_{dec}}C_0$$

Finally, since $\hat{L} = O(\kappa^2)$, $\bar{L} = O(\kappa^3)$ and $\mu\gamma = O(\kappa^{-1})$. Suppose we choose $I = O(1)$, then $C_\eta = O(\kappa^2)$, $C_\gamma = O(\kappa^5)$, and use $G_1 = \kappa(1-\tau\mu)^{Q+1}C_f$ in Proposition C.3, we have:

$$\frac{1}{T}\sum_{t=1}^{T}\left(\mathbb{E}\|\nabla h(x_t)\|^2 + \frac{1}{2I}\sum_{i=1}^{I}\mathbb{E}\|\mathbb{E}_\xi[\bar{\Delta}\hat{x}_{t,i}]\|^2\right)$$

$$= O\left(\frac{\kappa^3}{T} + \left(\frac{\kappa^5}{T}\right)^{1/2} + \left(\frac{\kappa^6}{T^2}\right)^{1/3} + \kappa^2(1-\tau\mu)^{2(Q+1)} + \kappa^4(1-\mu\gamma)^{I_{dec}}\right)$$

and to reach an $\epsilon$ stationary point, we choose $Q = O(\kappa\log(\frac{\kappa}{\epsilon}))$, $I_{dec} = O(\kappa\log(\frac{\kappa}{\epsilon}))$ and $T = O(\kappa^5\epsilon^{-2})$ number of iterations. $\square$