# OpenReview forum: "Resolving the Tug-of-War: A Separation of Communication and Learning in Federated Learning"
_NeurIPS.cc/2023/Conference — NeurIPS 2023 poster_

### Official Review · Reviewer_A2g4 · 2023-06-28

**Soundness:** 3 good
**Presentation:** 3 good
**Contribution:** 4 excellent
**Rating:** 7
**Confidence:** 4

**Summary:**

This paper proposes a two-layer federated learning (FL) framework (FedSep) by separating the communication and learning parameters. By a bilevel optimization formulation, FedSep enjoys a convergence guarantee. In addition, two settings, communication-efficient FL and model-heterogeneous FL, are solved in FedSep framework.

**Strengths:**

1. FedSep is an interesting and novel setting.
2. Theoretical convergence analysis is provided, showing the sublinear rate.
3. FedSep is applied to communication-efficient FL and model-heterogeneous FL.

**Weaknesses:**

I have the following concerns and questions:

1. The convergence rate is sublinear, but it does not show any speedup in terms of the clients' number and local steps. So it does not exactly match the convergence of standard FL algorithms. Any possible improvement? Or the statement in the paper should be modified accordingly (e.g., line 75).

2. In the model-heterogeneous FL, the validation data set is also used in the training process both for problem formulation and experiments. I believe this is a concern and unfair for the direct comparison with other algorithms

**Questions:**

see above

---

> ### Author Rebuttal · Authors · 2023-08-10
>
> Thanks for your positive comment. Below are responses to your questions:
>
> **Q1: is it possible to show speedup in terms of the clients' number and local steps**
>
> **A**: In Corollary 3.8, we focus on achieving the sub-linear rate. Indeed, it is possible to achieve the linear speedup in terms of the number of clients $M$ and the number of local steps $I$. More specifically, compared to corollary 3.8, we can set $\gamma$ to be a smaller value: $\gamma = \min(\frac{1}{2L},(\frac{1}{C_{\gamma}MIT})^{1/2})$,
> then we have
> a convergence bound of $O(\frac{1}{(MIT)^{1/2}})$,
> which achieves linear speed up w.r.t. $M$ and $I$. Recall that $\gamma$ denotes the learning rate used by the decode stage (Line 7 -12 in Algorithm~1), smaller $\gamma$ leads to more decode steps (larger $I_{dec}$). But since we assume the lower optimization problem $g$ be strongly convex (thus linear convergence), the additional cost is negligible.
>
> **Q2 In the model-heterogeneous FL, the validation data set is also used in the training process both for problem formulation and experiments. I believe this is a concern and unfair for the direct comparison with other algorithms.**
>
> **A**: We want to clarify that all methods (both our FedSep and other baselines) are trained over the same training set. While for our FedSep, we use a small subset of the training data to do the sub-net selection. In other words, the $D_{val}$ in Eq.6 is a subset of the training set in experiments. Therefore, it is a fair comparison with other baselines.

---

> > ### Comment · Reviewer_A2g4 · 2023-08-14
> >
> > I thank the authors for their detailed response. I have the concerns for the convergence. As suggested in Theorem 3.7 and Corollary 3.8, the algorithm can only go to an error ball with corresponding sublinear rate rather than achieving a stationary point convergence. The authors mentioned that the algorithm can achieve the linear speedup by choosing different learning rates. So what is the best rate it could actually achieve and what is the setting of the hyper-paremeters?

---

> > > ### Author Response · Authors · 2023-08-14
> > >
> > > Thanks for your reply. In fact, there is a **trade-off** between achieving linear-speed up w.r.t. $I$ and $M$ over controlling the error ball size. We add more details below:
> > >
> > > As shown in Line 624 of Appendix, we have the following bound before choosing a specific form of learning rate:
> > >
> > > $$ \frac{4\bar{L}h(x_{1})}{T\eta} + \frac{\eta G_2^2}{2Ib_xM} + C_{\eta}I^2\eta^2 + C_{\gamma}\gamma + 2G_1^2  + 2(3\kappa^2L^2 + \hat{L}^2)(1 - \mu\gamma)^{I_{dec}}C_0
> > > $$
> > >
> > > **In the original  Theorem 3.7**, we choose the learning rates as followings:
> > >
> > > $$
> > > \eta = \min\bigg(1, \bigg(\frac{8Ib_xM\bar{L}h(x_{1})}{TG_2^2}\bigg)^{1/2} ,\left(\frac{4\bar{L}h(x_{1})}{C_\eta I^2T}\right)^{1/3}\bigg),\; \gamma = \min\bigg(\frac{1}{2L}, \bigg(\frac{1}{C_{\gamma}T}\bigg)^{1/2}\bigg)
> > > $$
> > > which leads to a bound in the form of:
> > > $$
> > > \frac{4\bar{L}h(x_{1})}{T} + \bigg(\frac{2G_2^2\bar{L}h(x_{1})}{Ib_xMT}\bigg)^{1/2} + \left(\frac{16I^2C_{\eta}\bar{L}^2h(x_{1})^2}{T^2}\right)^{1/3}  + \bigg(\frac{C_{\gamma}}{T}\bigg)^{1/2} + 2G_1^2  + 2(3\kappa^2L^2 + \hat{L}^2)(1 - \mu\gamma)^{I_{dec}}C_0
> > > $$
> > > As shown by the bound above, the second and the fourth term are the dominant sub-linear terms, i.e. $\bigg(\frac{2G_2^2\bar{L}h(x_{1})}{Ib_xMT}\bigg)^{1/2} + \bigg(\frac{C_{\gamma}}{T}\bigg)^{1/2}$, where the term $\bigg(\frac{2G_2^2\bar{L}h(x_{1})}{Ib_xMT}\bigg)^{1/2} $ indeed has linear-speed up w.r.t $M$ and $I$, the bottle-neck is $\bigg(\frac{C_{\gamma}}{T}\bigg)^{1/2}$, which only scales w.r.t. $T$.
> > >
> > > **To achieve a linear speed up**, we instead reduce the value of $\gamma$ (step size of the decode problem) and choose the learning rates as follows:
> > >
> > > $$
> > > \eta = \min\bigg(1, \bigg(\frac{8Ib_xM\bar{L}h(x_{1})}{TG_2^2}\bigg)^{1/2} ,\left(\frac{4\bar{L}h(x_{1})}{C_\eta I^2T}\right)^{1/3}\bigg),\; \gamma = \min\bigg(\frac{1}{2L}, \bigg(\frac{1}{C_{\gamma}IMT}\bigg)^{1/2}\bigg)
> > > $$
> > >
> > > Then the dominant sublinear-term is in the form of  $\bigg(\frac{2G_2^2\bar{L}h(x_{1})}{Ib_xMT}\bigg)^{1/2} + \bigg(\frac{C_{\gamma}}{IMT}\bigg)^{1/2}$, **thus has a linear speed up w.r.t. both $I$ and $M$**. Meanwhile, note that the error ball term is $2G_1^2  + 2(3\kappa^2L^2 + \hat{L}^2)(1 - \mu\gamma)^{I_{dec}}C_0$, where the second term also depends on $\gamma$, as a result, we then need to choose a large value of $I_{dec}$ to control the value of the error ball, but since the size of the error ball depends on $I_{dec}$ exponentially, we can choose small value of $I_{dec}$ to get satisfactory convergence in practice.
> > >
> > > Please let us know if you have any further concerns.

---

> > > > ### Comment · Reviewer_A2g4 · 2023-08-14
> > > >
> > > > Thanks for the response. I am curious about the error term $G_1$? It is independent of hyper-parameters' settings. So it will always remain constant and thus the error ball is irreducible in such sense.

---

> > > > > ### Author Response · Authors · 2023-08-14
> > > > >
> > > > > Thanks for your question again.
> > > > >
> > > > > In fact, $G_1$ is not a constant term. As shown by Proposition C.3 in the manuscript, we have $G_1 = \kappa(1 - \tau\mu)^{Q+1}C_f$, and recall that $Q$ is the number of **Encode steps** as defined in the Eq. 3 in the manuscript. As a result, to control this error for reaching an $\epsilon$ stationary point, we can choose $Q = O(\kappa\log(\frac{\kappa}{\epsilon}))$, which has a logarithm dependency over $\epsilon$, thus negligible in practice. This is discussed in the Theorem C.8 of appendix, and we omit the details of $G_1$ in Theorem 3.7 for clarity. Sorry for the confusion.

---

### Official Review · Reviewer_G158 · 2023-07-04

**Soundness:** 4 excellent
**Presentation:** 4 excellent
**Contribution:** 4 excellent
**Rating:** 6
**Confidence:** 3

**Summary:**

The authors proposed a two-layer federated learning framework called FedSep, with one layer for communication and another layer for learning. The two layers are connected through decode/encode operations. Furthermore, the authors proposed an efficient algorithm to solve FedSep by treating it as a bilevel optimization problem, and showed convergence results which match those of the standard framework. The FedSep can incorporate Communication-Efficient FL and Heterogeneous-Model FL.

**Strengths:**

(S1) The proposed framework, FedSep, provided a general framework for resolving the tug-of-war by separating the communication and learning of federated learning into two layers.

(S2) The algorithm was explained in detail and the theoretical results are solid. The experiments results supports the claims of the authors well.

**Weaknesses:**

(W) The framework rely on the analyticity and strong convexity of the second level problem, which may not be true in many cases. For example, for problem (equation 4) formulated in Sec 4.1, the analyticity and strong convexity does not seem to hold. Although I still believe some convergence results can be derived.

**Questions:**

(Q) The authors claimed that Theorem 3.7 implies that $\mathbb{E}||\nabla h(x_t)||^2 + \frac{1}{2I}\sum_{i=1}^I \mathbb{E}||\mathbb{E}_{\xi}[\bar{\Delta} \hat{x}_t, i ]||$ converges to zero. Because the right-hand side of the inequality in Theorem 3.7 contains constant, this claim is not straightforward to me, can the authors elaborate?

---

> ### Author Rebuttal · Authors · 2023-08-10
>
> Thanks for your positive comment. Below are responses to your concerns and questions:
>
> **W: The framework rely on the analyticity and strong convexity of the second level problem.**
>
> **A**: In the bilevel optimization literature, the Non-convex-strongly-convex assumption (including the smoothness assumptions) is widely used. These assumptions ease the analysis of the bilevel optimization problem: strong convexity of the lower problem makes the hyper-gradient well-defined, as an analytical form of the hyper-gradient requires the Hessian to be positive definite.  As for the more general non-convex lower level optimization problem, it is much more challenging to analyze, and we leave the analysis of this case as a future work
>
>
> **Q: how the left-hand side of Theorem 3.7 converges to 0?**
>
> **A**: One approach to achieve the convergence is choosing diminishing learning rate as stated in Corollary 3.8. In our updated version of the manuscript, we will move this discussion after the Corollary 3.8 to avoid confusion.

---

> > ### Comment · Reviewer_G158 · 2023-08-14
> >
> > I thank the authors for their response. But I still have two questions:
> >
> > (1) the author does not answer why the (equation 4) formulated in Sec 4.1 satisfies the analyticity and strong convexity.
> >
> > (2) The are still constants on the right-hand-side of Corollary 3.8.
> >
> > Therefore, I choose to keep my scores.

---

> > > ### Author Response · Authors · 2023-08-20
> > >
> > > Thanks for your comment.
> > >
> > > **Q1**:  Eq. 4 has an $L_1$ regularization term, thus it is not smooth nor strongly convex.  So we cannot use the convergence Theorem 3.7 to predict its convergence. However, we can still use the proximal gradient descent algorithm to perform the decode operation and Eq.5 to perform the encode operation. Empirically (over MNIST and CIFAR-10), we get good performance. As stated in our previous response, the general non-smooth non-strongly-convex lower level optimization problem is much more challenging to analyze, and is an ongoing research topic in bilevel optimization research. We leave the analysis of this case as a future work.
> > >
> > > **Q2**: Please refer to the answer to Question 3 in the overall response for the trade-off between encode/decode error and the overall convergence. Furthermore, the constant term $G_1^2 + \kappa^4(1 - \mu\gamma)^{I_{dec}} = \kappa(1 - \tau\mu)^{Q+1}C_f + \kappa^4(1 - \mu\gamma)^{I_{dec}} $, has a logarithm dependence over $Q$ and $I_{dec}$ thus is negligible in pracice.

---

### Official Review · Reviewer_Dwjg · 2023-07-06

**Soundness:** 3 good
**Presentation:** 2 fair
**Contribution:** 2 fair
**Rating:** 3
**Confidence:** 3

**Summary:**

This paper asserts that the tasks of communication and learning are at odds in federated learning. As such, a new approach is suggested that separates these tasks. The approach comprises an encode-decode operation, where decoding is cast as an optimization problem. The overall learning task is formulated as a bilevel optimization problem. Standard first-order gradient-based approaches are then employed to solve the bilevel optimization problem. The structure of the proposed algorithm mirrors that of typical FL algorithms. A convergence bound is proven for this algorithm. Some applications pertaining to communication-efficiency and model-heterogeneity in FL are also discussed.

**Strengths:**

- The particular "separation" framework suggested in this paper appears to be novel; I have not seen anything exactly like this in earlier FL work.

- One advantage of the proposed bilevel approach is that each agent can train a separate local model $y^{(m)}$ of its own. This allows for personalization in the face of data-heterogeneity.

**Weaknesses:**

I have several major concerns ranging from the motivation to the utility of the proposed approach. Let me elaborate on them below.

- The entire paper is based on the premise that communication and learning are conflicting goals in federated learning. However, this statement is never formalized at any point in the paper. The only discussion regarding the tension between communication and learning shows up in lines 25-30, which don't convey anything concrete.

- Continuing with the above point, I also missed the motivation for the specific bilevel formulation. By now, there are several approaches that guarantee communication-efficiency in RL (see Refs [R1]-[R5] below) using techniques ranging from quantization, sparsification, compressed sensing, etc. It stands to reason that compressing information about models/gradients will naturally come at the cost of performance in learning. So the tension here is evident, and has been well-explored/quantified in the papers I mentioned earlier. In particular, sending less bits of information slackens the rate of convergence, and one can potentially try to investigate the fundamental limits on the rate needed to achieve a desired level of accuracy; see, for instance, [R5]. However, there is no clear discussion of why the FedSep bilevel idea is any significant improvement over any of these MANY existing schemes.

[R1] Federated learning with compression: Unified analysis and sharp guarantees, Haddadpour et al., AISTATS 21

[R2] Linear convergence in federated learning: Tackling client heterogeneity and sparse gradients, Mitra et al., NeuRIPS 21

[R3] Optimizing the communication-accuracy trade-off in federated learning with rate-distortion theory, Mitchell et al., arXiv 2022

[R4] Communication-Efficient Federated Learning through Importance Sampling, Isik et al., arXiv 2023

[R5] Differentially quantized gradient methods, Yin et al., IEEE Transactions on Information Theory, 2022

- Regarding model-heterogeneity, some of the references I alluded to earlier do account for heterogeneous loss functions across agents. Moreover, if the goal is to allow for personalized models, then why not use any of the existing schemes for personalization in FL (of which, there are many)? See, for instance, ideas based on Moreau Envelopes and MAML in [R6] and [R7], and representation learning in [R8]. If one cares about both personalization and communication-efficiency, I can imagine simply using these ideas in tandem (in meaningful ways).

[R6] Personalized Federated Learning with Moreau Envelopes, Dinh et al, NeuRIPS 2020

[R7] Personalized Federated Learning: A Meta-Learning Approach, Fallah et al., NeuRIPS 2020

[R8] Exploiting Shared Representations for Personalized Federated Learning, Collins et al, ICML 21

The overarching point I am trying to make here is that each of the key considerations in FL that the authors allude to (communication-efficiency, heterogeneity, personalization, etc), have several existing principled algorithmic solutions. It wasn't apparent to me at all why there is a compelling need to depart from these existing approaches.

- In addition to the motivation, several parts of the paper are somewhat vaguely written. For instance, in Eq. (1), the meaning of the object $g^{(m)}$ is not explained. The encoding operator in Eq.~(2) isn't clear to me either. What is this operation and how does it compare with any of the standard encoding techniques (say for instance, standard scalar and vector quantizers, or sparsifying mechanisms like Top-k)? How many bits are needed to perform this encoding? What is the error caused by this encoding? Is this an unbiased encoding mechanism? No intuition is provided at all about any of these crucial points, making it hard to draw meaningful comparisons with existing schemes.

- The difference with existing federated bilevel optimization algorithms in lines 153-157 is also quite terse. The discussion did not come across as anything fundamental. It wasn't clear to me why the bilevel algorithm proposed can't be analyzed by simply adapting ideas from other existing FL bilevel algorithms.

- The main convergence result in Theorem 3.7 is hard to parse. I was left wondering about several key questions: (i) How does the compression scheme (encoding-decoding) affect the rate of convergence? (ii) How does this trade-off compare with the existing known bounds for compression in FL? (iii) How does the effect of heterogeneity in the agents' loss functions manifest in the bounds? Does this dependence match with those known for federated bilevel optimization?

Also, the iteration complexity seems to have a $\kappa^5$ dependence on the condition number $\kappa$. This seems much larger than what one typically obtains. Thus, I am not convinced about the tightness of the bounds either.




**Questions:**

No questions other than the ones above.

**Limitations:**

I couldn't find a clear discussion of the limitations.

---

> ### Author Rebuttal · Authors · 2023-08-10
>
> Thanks for spending time reviewing our paper. Below are our responses to your questions:
>
> **A(W1: Formal discussion of communication/learning conflict)**: Please refer to the answer for **Q1 in the overall response.**
>
> **A(W2: Comparison with existing compression FL methods)**: Please refer to the answer for **Q2 in the overall response.**
>
> **A(W3: Comparison with existing personalized FL models)**: Firstly, FedSep provides a general framework to deal with the communication-learning dilemma in FL. [1]-[3] can be viewed as a special case of FedSep when we choose proper decode functions: For the Moreau Envelopes based Personalized FL [1], we set both the decode optimization problem ($h^{(m)}(x)$) and the local training objective ($f^{(m)}(x)$) as a regularized loss function with l2-norm; for the MAML-based method [2], we set the decode problem as one step gradient descent; while the representation learning [3] method corresponds to a degenerated case where the communication parameter (global representation) and learning parameter (client-specific heads) are not connected through a decode problem. Note that [1]-[3] are designed for the data heterogeneity of FL, our FedSep can also incorporate the model-heterogeneity FL as discussed in the Section 4.2. In model-heterogeneous FL, we can choose different scale of models based on the available resources of each individual client.
>
> Next, although various aspects of the communication-learning dilemma have been studied in the literature, these methods are proposed in **isolation**. FedSep provides the first unified analysis framework for them using the theory of bilevel optimization. Furthermore, our FedSep can be used to design novel algorithms  as demonstrated by the communication-efficient algorithm and model-heterogenous algorithm in Section 4 of the manuscript.
>
> **A(W4: Discussion of the encoder operation)**: $g^{(m)}$ is the minimization problem solved in the decode operation by the client m;
>
> About the encode operation Eq.2, it is the **inverse operation of the decode operation**. Recall that the decode operation is defined as $y^{(m)}_x = Dec(x)$, where $y^{(m)}_x$ is the minimizer of $g^{(m)}(\cdot, x)$. For the encode operation, it measures how the communication parameter is changed when the minimizer $y^{(m)}_x$ changes, i.e. $\nabla_x y^{(m)}_x = Enc(\cdot)$. Eq.2 shows an explicit form of $\nabla_x y^{(m)}_x$ and can be derived following the implicit function theorem (a standard result in bilevel optimization). In section 4.1, we show an instantiation of the encode/decode operations in the context of communication-efficient FL: the decode function is a LASSO problem that solves the high-dimensional learning parameter from a low dimensional communication parameter with a sparse constraint, while the encode operation is a non-linear mapping as shown in Eq.5 in the manuscript. Finally, The dimension of the communication parameter can be chosen arbitrarily, however, we would see a coding-accuracy trade-off. Please refer to our answer to **Q3 in the overall response** for a more detailed analysis of this trade-off.
>
>
> **A(W5: Difference with existing federated bilevel optimization algorithms)**: Our Algorithm 1 is much more efficient compared to a naive application of the bilevel optimization algorithm. Suppose we use a standard bilevel optimization algorithm to solve FedSep, then we perform multiple rounds of the following steps 1-4: step 1: we decode the communication parameter to get the learning parameter (solve the inner optimization problem); step 2: we compute the gradient of the learning objective w.r.t. the learning parameter; step 3: we compute the hyper-gradient w.r.t. the communication parameter through the encode operation; step 4: we use gradient descent to update the communication parameter.
>
> This algorithm is actually a direct application of the FedAvg to a bilevel optimization problem, which is inefficient due to performing the decode (step 1) and encode (step 3) operations multiple times. **Our idea is to optimize the learning objective at the learning parameter space**. As shown by Algorithm 1, we perform the following steps: step 1: same as the naive approach; step 2: we perform multiple gradient descent steps to optimize learning objective at the learning parameter space; step 3: we map the learning parameter update back to the communication parameter space through the encode operation; step 4: same as the naive approach. **Compared to the naive approach, we only perform decode/encode operation once.**
>
> The convergence analysis is also challenging. If we follow the naive approach, we can get accurate estimation of the hyper-gradient (gradient w.r.t to the communication parameter), and the convergence analysis is a straightforward application of FedAvg analysis. However, since we perform multiple gradient descent steps at the learning parameter space, we would have a biased hyper-gradient estimation. Then we need to bound this hyper-gradient estimation error carefully (see Lemma C.6 for more details). Finally, in corollary 3.8, we show that the communication parameter $x$ reaches to a stationary point of the bilevel problem (in Eq.1), and the learning parameter $y^{(m)}$ converges to the stationary point of the local learning problem $f^{(m)}(y)$.
>
> **A(W6: Discussion of the convergence results)**: Please refer to the answer for **Q3** in the overall response.
>
> **References**
>
> [1] Personalized Federated Learning with Moreau Envelopes, Dinh et al, NeuRIPS 2020
>
> [2] Personalized Federated Learning: A Meta-Learning Approach, Fallah et al., NeuRIPS 2020
>
> [3] Exploiting Shared Representations for Personalized Federated Learning, Collins et al, ICML 21

---

> > ### Author Response · Authors · 2023-08-11
> > **Further Discussion over the Accuracy-Coding Trade-off**
> >
> > In the answer to **Q3 of the overall response**, we discuss the trade-off between coding and accuracy, we add more details about it here.
> >
> > For the **decode** operation, we solve the minimization problem $\underset{y^{(m)} \in \mathbb{R}^{d^{(m)}}}{\arg\min} g^{(m)}(x, y^{(m)})$, while in Algorithm 1 (Line 7-12), we solve this problem with $I_{dec}$ number of steps, therefore, we have a bias term at the order of $(1 - \mu\gamma)^{I_{dec}}$ for the decode operation. Naturally, we can increase $I_{dec}$ to reduce the bias of the decode operation. Then for the **encode** operation, we map the update of the learning parameter back to the communication parameter space following Eq.2. In practice, we evaluate Eq.2 approximately using Eq.3, and Eq.3 is a biased estimation of Eq.2, where we have the variance term and bias term denoted as $G_1$ and $G_2$ respectively. As shown by Proposition 3.5, we can reduce the bias $G_1$ by increasing the value $Q$ in Eq.3 and reduce the variance term $G_2$ by increasing the batch-size $b_x$ (the size of the mini-batch $B_x$  in Eq.3).
> >
> > The effects of these bias/variance terms are illustrated in Theorem 3.7 (Corollary 3.8). A convergence bound with explicit dependence over $G_1$, $G_2$ and $(1 - \mu\gamma)^{I_{dec}}$ is $O\big(\frac{G_2}{\sqrt{T}}+ \tilde{G}\big)$, where $ \tilde{G} = (1 - \mu\gamma)^{I_{dec}} + G_1$ is the sum of the decoder bias and encoder bias. As shown by this bound, both the bias and variance terms affect the convergence of the algorithm. Furthermore, the bias term decrease exponentially  w.r.t. $I_{dec}$ and $Q$, thus small values leads to satisfactory performance in practice. As for the variance term $G_2$, it decrease linearly w.r..t the batch-size $b_x$, which has the same dependence of stochastic noise in FedAvg.

---

> > ### Comment · Reviewer_Dwjg · 2023-08-15
> > **Thank you for your response**
> >
> > I thank the authors for their rebuttal. One of my major concerns about the paper -  which I explained in my first set of comments - is the need for a new bilevel framework, given the existence of several communication-efficient algorithms for optimization. Moreover, I asked how the new FedSep approach compares to the already known bounds for such algorithms.
> >
> > I agree that the FedSep approach offers a potentially novel perspective to encoding-decoding information in FL. However, it is still not clear to me whether the performance of FedSep is any better than what is already known. Let me elaborate.
> >
> > - Consider the paper "Differentially quantized gradient methods, Yin et al., IEEE Transactions on Information Theory, 2022" that I pointed to in my initial set of comments. For smooth and strongly convex optimization (without noise), this paper proposes a quantized gradient-descent algorithm that can retain the exact same linear convergence rate as without quantization, when sufficient bits are used for encoding. The number of such bits is also shown to be tight in this work. So for deterministic optimization, this paper provides a clear benchmark for comparison.
> >
> > - Similarly, for stochastic optimization, the paper "The error-feedback framework: Better rates for SGD with delayed gradients and compressed communication" shows that by using error-feedback, one can retain nearly the same rates for sparsified SGD variants as without sparsification. Essentially, the effect of compression is relegated to higher order terms that are asymptotically negligible. Several subsequent variants have been developed for the multi-agent case.
> >
> > So in short, any claims of usefulness of FedSep in terms of communication-learning trade-offs should be carefully contrasted with the results in these papers. Without a precise comparison revealing that FedSep performs no worse than these other schemes, I remain unconvinced of its utility.
> >
> > - In terms of personalization, I am a bit confused. The authors mentioned that all the prior approaches for personalization using Moreau envelopes and MAML can be encompassed within their broader unified framework. If that is the case, then the main results in these papers should fall out as special cases of the main convergence result for FedSep. Can the authors argue that this is indeed the case?
> >
> > For instance, consider the two papers below:
> >
> > [1] Personalized Federated Learning with Moreau Envelopes, Dinh et al, NeuRIPS 2020
> >
> > [2] Personalized Federated Learning: A Meta-Learning Approach, Fallah et al., NeuRIPS 2020
> >
> > For suitably chosen encoding-decoding functions, can the authors show explicitly that the results in the above papers are special cases of Theorem 3.7.?

---

> > > ### Author Response · Authors · 2023-08-21
> > >
> > > Thanks for your reply. Below is our response to your further comments.
> > >
> > > **Comparison with Communication-efficient FL works**. Since we study the convergence of non-convex functions, we can compare with the results in the second paper [3] you mentioned.  More specifically, the rate achieved by [3] is $O(\frac{1}{\delta T} + \frac{\sigma}{\sqrt{T}})$, where $\delta$ is the bias of compressor and $\sigma$ is the variance of stochastic gradient estimation. As for our FedSep, we have convergence rate of $O(\frac{G_2}{\sqrt{T}} + G_1^2 + \kappa^4(1 - \mu\gamma)^{I_{dec}})$, where $G_2 = O(\frac{1}{b_x})$ ($b_x$ is the batch-size, and see Proposition C.3 for detailed expression) is the variance term related to encode operation, $G_1= \kappa(1 - \tau\mu)^{Q+1}C_f$ is the bias term related to the encode operation and $\kappa^4(1 - \mu\gamma)^{I_{dec}}$ is bias term related to the decode operation. So, our FedSep gets **NO WORSE RESULTS** than in [3]: the bias term does not show up in the dominant term,  and only the variance appears in the dominant  term.  **Meanwhile, our FedSep does not involve any error-feedback steps, and thus is stateless (a desired property for FL)**
> > >
> > >
> > > **Comparison with other Personalized FL works**. We make a thorough comparison between our FedSep and the pFedMe [1] framework below.  *As for the Per-FedAvg[2], as argued in Eq. 5 of [1], the problem formulation studied by [2] considers a first-order approximation of the problem studied in [1], so the comparison can be extended to Per-FedAvg straightforwardly.*
> > >
> > > **Compare with pFedMe**[1]:
> > > * **Formulation**: Our FedSep **recovers** the formulation of pFedMe by setting the communication parameter as $\omega$, the learning parameter as $\theta_i$ on the $i_{th}$ client,  the decoding function (the lower level problem g in our framework, Eq. 1) as $f_i(\theta_i)  +\frac{\lambda}{2}\|\theta_i - \omega\|^2$ and the learning problem (the upper level problem f in our framework, Eq. 1) $f_i(\theta_i(\omega))  +\frac{\lambda}{2}\|\theta_i(\omega) - \omega\|^2$, where $\theta_i(\omega)$ is the minimizer of the decode problem.  A special characteristic of this formulation is that its decode problem and learning problem are the **SAME**.
> > >
> > > * **Algorithm**: Translate Algorithm 1 into the language of our framework, it performs multiple rounds of decode (solving the Moreau Envelop problem), encode(Compute the hyper-gradient of the learning problem w.r.t. $\omega$, i.e. $\lambda(\theta_i(\omega) - \omega)$) operations per global round, due to the special structure of its formulation: the learning problem is the same as the decode problem, there is no need to solve the learning problem after the decode operation. In comparison, our algorithm is designed for the general setting (the learning problem is the different from the decode problem), and performs the decode/encode operation once and solve the learning problem with multiple steps. For the case where encode/decode operations are expensive, such as in the communication-efficient FL, our algorithm is more computation-efficient.
> > >
> > > * **Convergence Result**: Due to the differences between algorithm and specific details of analysis, our convergence theorems are not exactly the same. However, there are some important **observations**. **First**, the authors claim (in Theorem 2 of [1]) that pFedMe algorithm has convergence rate of $1/(TRN)^{2/3}$ which outperforms the standard single level FL algorithm, but it also has an **irreducible constant noise term $\sigma_{F,2}^2/\lambda^2$**, in contrast, we recover the convergence rate of standard single level FL algorithm and does not have this constant noise term. **Second**, note that in Theorem 2 of [1], its has an error ball of size $O(\delta^2)$ (error of solving Moreau Envelop problem), this corresponds to the the bias term of decode operation $\kappa^4(1 - \mu\gamma)^{I_{dec}}$ in Theorem 3.7; due to the special formulation (learning problem is the same as decode problem in pFedMe), the encode operation is unbiased, thus Theorem 2 of [1] does not have a dependence over the bias of encode operation. For the more general setting where the learning and decode problem is not the same, we show in our Theorem 3.7 that there is also an additive dependence over encode operation bias $G_1$.
> > >
> > >
> > > **References**
> > >
> > > [1] Personalized Federated Learning with Moreau Envelopes, Dinh et al, NeuRIPS 2020
> > >
> > > [2] Personalized Federated Learning: A Meta-Learning Approach, Fallah et al., NeuRIPS 2020
> > >
> > > [3] Stich, Sebastian U., and Sai Praneeth Karimireddy. "The error-feedback framework: Better rates for SGD with delayed gradients and compressed communication." arXiv preprint arXiv:1909.05350 (2019).

---

> > > > ### Author Response · Authors · 2023-08-21
> > > >
> > > > Thanks for your engagement into the discussion again. We made a few narrative adjustments in our most recent response to your questions about comparison with communication-efficient FL with error-feedback and personalized FL methods. Please let use know if there are any further clarifications that we can make.

---

### Official Review · Reviewer_WQf3 · 2023-07-12

**Soundness:** 3 good
**Presentation:** 3 good
**Contribution:** 3 good
**Rating:** 6
**Confidence:** 4

**Summary:**

The paper uses bilevel optimization in a federated learning setting. A unique decomposition of communication and learning has been identified, which has applications for reducing communication overhead and supporting heterogeneous models. The theoretical convergence guarantee has been presented and experiments also show the advantage of the proposed method.

**Strengths:**

- The use of bilevel optimization as a decomposition into communication and learning in a FL setting is interesting.
- The framework is general and supports multiple application scenarios of FL.
- Theoretical convergence bound is provided.

**Weaknesses:**

- The use of encoder/decoder structure may incur additional computation compared to common FL algorithms. It would be nice to quantify the amount of such additional computation.
- The extension of common bilevel optimization algorithms to include multiple update steps (as pointed out at the end of page 4) seems to be somewhat similar to the idea of local updates in FedAvg / local SGD algorithms. It would be nice to identify what are the key technical challenges and novel solution techniques in this extension.
- There is space for improvement in the experiments (see details below).
- The writing could be improved to highlight the usefulness of encoder/decoder in the context of FL, in early parts of the paper.

**Questions:**

- What is the additional computation overhead compared to standard FL algorithms such as FedAvg?
- What are the key technical (mathematical) challenges and novelties?
- In the experiments in Section 5.1, biased compressors such as top-K usually only works well with error feedback [a]. Is error feedback used in the baseline methods? In addition, are the first and third plots in Figure 2 results at convergence? Theoretically, all methods (when using error feedback in baselines) should converge to the optimal solution when the experiments are run long enough. More explanation is needed on why a much worse accuracy is obtained when the compression rate is high. Is it a matter of hyper parameter tuning? It may be better to plot the accuracy against the amount of communication (e.g., in the number of bytes transmitted, *not* in communication rounds), to compare the accuracies of different methods and different compression rates at the same amount of communication.
- In Section 5.2, the advantage of having heterogeneous models is not quite clear.

[a] Stich, Sebastian U., and Sai Praneeth Karimireddy. "The error-feedback framework: Better rates for SGD with delayed gradients and compressed communication." arXiv preprint arXiv:1909.05350 (2019).

---

> ### Author Rebuttal · Authors · 2023-08-10
>
> Thanks for spending time reviewing our paper. Below are our responses to your questions:
>
> Response to Concerns:
>
> **A(W1: Additional computational cost)**: Compared to FedAvg, FedSep needs to perform extra decode and encode operations. For the decode operation, we need to perform $I_{dec}$ of gradient descent steps to optimize the decode problem (Line 7-12 of Algorithm 1), while for the encode operation, we need to perform $Q$ Hessian-vector products (Eq.3). Since the estimation error of both decode and encode operations have linear convergence, we can choose small values of $I_{dec}$ and $Q$ in practice, thus is negligible. Furthermore, to incorporate the communication-learning trade-offs, most existing approaches needs some extra encode/decode operations. For example, FetchSGD [1] is a communication-efficient FL method and uses Count-sketch to compress the local updates before communicating with the server.
>
> **A(W2: Technical novelty and difficulty)**: Our Algorithm 1 is much more efficient compared to a naive application of the bilevel optimization algorithm. Suppose we use a standard bilevel optimization algorithm to solve FedSep, then we perform multiple rounds of the following steps 1-4: step 1: we decode the communication parameter to get the learning parameter (solve the inner optimization problem); step 2: we compute the gradient of the learning objective w.r.t. the learning parameter; step 3: we compute the hyper-gradient w.r.t. the communication parameter through the encode operation; step 4: we use gradient descent to update the communication parameter.
>
> This algorithm is actually a direct application of the FedAvg to a bilevel optimization problem, which is inefficient due to performing the decode (step 1) and encode (step 3) operations multiple times. **Our idea is to optimize the learning objective at the learning parameter space**. As shown by Algorithm 1, we perform the following steps: step 1: same as the naive approach; step 2: we perform multiple gradient descent steps to optimize learning objective at the learning parameter space; step 3: we map the learning parameter update back to the communication parameter space through the encode operation; step 4: same as the naive approach. **Compared to the naive approach, we only perform decode/encode operation once.**
>
> The convergence analysis is also challenging. If we follow the naive approach, we can get accurate estimation of the hyper-gradient (gradient w.r.t to the communication parameter), and the convergence analysis is a straightforward application of FedAvg analysis. However, since we perform multiple gradient descent steps at the learning parameter space, we would have a biased hyper-gradient estimation. Then we need to bound this hyper-gradient estimation error carefully (see Lemma C.6 for more details). Finally, in corollary 3.8, we show that the communication parameter $x$ reaches to a stationary point of the bilevel problem (in Eq.1), and the learning parameter $y^{(m)}$ converges to the stationary point of the local learning problem $f^{(m)}(y)$.
>
> **A(W3: Experimental setting)**: See our response to **Q3** below.
>
> **A(W4: Usefulness of encoder/decode)**: The encoder/decoder are bridges to connect the communication parameter and learning parameter. Selecting the appropriate decode/encode operation is central to applying FedSep in addressing various challenges within Federated Learning (FL). As shown by the communication-efficient FL task and model-heterogenous task.
>
> Response to Questions:
>
> **A(Q1: Additional computational cost)**: Please see response to the **W1** above.
>
> **A(Q2: Technical novelty and difficulty)**: Please see response to the **W2** above.
>
> **A(Q3: Experimental setting)**:  We include the error feedback for Top-K method.
>
> We compare our FedSep with other baselines using the following schema in the first and third plot of Figure 2: First, we run the uncompressed baseline with a given number of global epochs (until convergence), and we denote its communication budget as $P$. Note that the uncompressed method has a compression rate of 1. **For other methods, we vary hyper-parameters of each method under a given compression rate and report the best accuracy**. Take the Top-K method as an example, under the compression rate 10, we run Top-K with different values of K until reaching the communication budget of $0.1P$, and report the best accuracy achieved.
> In summary, the compression rate is defined by the ratio between the amount of bits transmitted with the total amount of communication budget ($P$). By using the this comparison scheme, we can **compare the rate limit of each method under a given communication budget**, note that [1] also uses a similar scheme for comparison. Furthermore, by reading the first and third plot of Figure 2 from right to left, we can see that the accuracy indeed increases when the amount of communication increases.
>
> **A(Q4: Advantages of having heterogeneous models)**: Training heterogeneous models is used to incorporate the different resource limit of clients, some clients have the capacity to train large model, while other clients can only train a small model. In particular, we implement one way of achieving heterogeneous model with FedSep: data-driven sub-network selection. As shown by Table 1, our FedSep outperform other sub-network selection methods, and achieve comparable accuracy with the Homogeneous (large) baseline, where all clients train the full-size model.
>
>
>
> **References**
>
> [1] D. Rothchild, A. Panda, E. Ullah, N. Ivkin, I. Stoica, V. Braverman, J. Gonzalez, and R. Arora.453
> Fetchsgd: Communication-efficient federated learning with sketching. In International Confer-454
> ence on Machine Learning, pages 8253–8265. PMLR, 2020.

---

> > ### Author Response · Authors · 2023-08-11
> > **Additional Experiments: Accuracy w.r.t. The amount of Communication**
> >
> > In the second and the fourth plots of Figure 2, we show the Test accuracy w.r.t. the communication rounds when we transmit different number of parameters. As suggested by the reviewer, we also show the **Test accuracy w.r.t. the Total communication** for our FedSep below.
> >
> > Table 1 shows the results for the I.I.D case, while  Table 2 shows the case of Non. I.I.D. **Note** that we denote one communication unit as transferring bits that equals to the size of the full model. Each column represents a different amount of communication, and each row represents a different choice of the number of parameters transferred per round.
> >
> > **Table 1**: Test Accuracy w.r.t. the Total Amount of Communication (I.I.D)
> >
> > |  Num. Params. / Comm.     | 5      | 20     | 30     | 40     | 50     |
> > |-------|--------|--------|--------|--------|--------|
> > | 20000 | 0.513  | 0.905  | 0.933  | 0.951  | **0.969**  |
> > | 10000 | 0.434  | 0.914  | **0.945**  | **0.966**  | 0.967  |
> > | 5000  | 0.375  | **0.922**  | 0.941  | 0.956  | 0.961  |
> > | 2000  | 0.353  | 0.896  | 0.936  | 0.943  | 0.956  |
> > | 500   | **0.627**  | 0.878  | 0.880  | 0.886  | 0.886  |
> >
> >
> > **Table 2**: Test Accuracy w.r.t. the Total Amount of Communication (Non. I.I.D)
> >
> > |  Num. Params. / Comm.     | 5     | 20    | 30    | 40    | 50    |
> > |-------|-------|-------|-------|-------|-------|
> > | 20000 | 0.262 | 0.820 | 0.862 | 0.901 | 0.919 |
> > | 10000 | 0.183 | 0.847 | 0.886 | **0.917** | **0.923** |
> > | 5000  | 0.226 | **0.862** | **0.894** | 0.916 | 0.920 |
> > | 2000  | 0.217 | 0.811 | 0.848 | 0.862 | 0.881 |
> > | 500   | **0.356** | 0.696 | 0.754 | 0.731 | 0.740 |
> >
> > From the tables, we observe that: when given a low communication budget, we get the best accuracy by transferring smaller number of parameters (high parameter compression rate). As we increase the budget, we get better accuracy by transferring more parameters per round. **Intuitively**, Take more rounds of training tend to get a higher accuracy, while transferring smaller number of parameters leads to more rounds of training for a given amount of communication budget.

---

> > ### Comment · Reviewer_WQf3 · 2023-08-16
> > **Not clear whether error-feedback has been used in the top-K baseline**
> >
> > Thanks for the response. It remains unclear whether error-feedback has been used in the top-K baseline in the experiments. This is an important point, because top-K can have much better performance with error-feedback compared to without error-feedback, since it is a biased compressor. A related point has also been mentioned in Reviewer Dwjg's latest comment (on Aug. 15). There exist several papers on this topic. For example, a related work that applies error-feedback top-K on both the client and server is Tang et al., DoubleSqueeze: Parallel Stochastic Gradient Descent with Double-Pass Error-Compensated Compression, ICML 2019.
> >
> > I would suggest the authors to compare with error-feedback top-K, if the current top-K implementation in the baseline does not include error-feedback.

---

> > > ### Author Response · Authors · 2023-08-20
> > >
> > > Thanks for your comment! We carefully check the code implementation. For topK baseline, we use the standard error-feedback schema as shown in the Algorithm 1 of [1]. **However, we find that we add the error-feedback term to the gradient instead of the learning rate  times gradient**, (See Line 3 of Algorithm 1 of [1]) this is equivalent to add a shrinking coefficient (equals to the learning rate) to the error feedback, which weakens the benefit of using error-feedback. Indeed, its performance is close to the vanilla Top-K methods.
> > >
> > > In Table-1 and Table-2 below, we show the corrected Top-K results for I.I.D and Non I.I.D MNIST dataset (First and Third Plots in Figure 2). As shown by results, the accuracy improves a lot especially, for the Non I.I.D case. However, we still notice that Top-K diverges under high parameter compression rate. In particular, when we set $K=1500$, it diverges under both I.I.D and Non I.I.D settings. In contrast, our FedSep still  gets good performance with 71\% accuracy for the Non I.I.D setting after 2000 rounds.  In our final version, we will also correct the results for the CIFAR-10 dataset.
> > >
> > > Table 1: Accuracy vs Compression Rate for I.I.D MNIST
> > > |  Compression Rate    | 50    | 20    | 10    | 5     |
> > > |-------|-------|-------|-------|-------|
> > > |FedSep | **0.9438**| **0.9792**| 0.9820| 0.9848|
> > > |Count-Sketch | 0.9199| 0.9639| 0.9755| 0.9852|
> > > |Top-K   | 0.9354| 0.9667| **0.9888**| **0.9875**|
> > >
> > > Table 2: Accuracy vs Compression Rate for Non. I.I.D MNIST
> > > |   Compression Rate    | 50    | 20    | 10    | 5     |
> > > |-------|-------|-------|-------|-------|
> > > |FedSep | **0.8930**| **0.9318**| 0.9337| 0.9044|
> > > |Count-Sketch | 0.5966| 0.9186| 0.9413| **0.9659**|
> > > |Top-K   | 0.8810| 0.9235| **0.9443**| 0.9428|
> > >
> > > References
> > >
> > > [1] Stich, Sebastian U., Jean-Baptiste Cordonnier, and Martin Jaggi. "Sparsified SGD with memory." Advances in Neural Information Processing Systems 31 (2018).

---

> > > > ### Comment · Reviewer_WQf3 · 2023-08-21
> > > >
> > > > Thanks for the update! In theory, top-K with error feedback should not diverge when proper learning rates are chosen, so I recommend the authors to do a grid search of learning rates for *each algorithm*, to ensure that the (approximately) optimal learning rates are used for each algorithm, similar to what's done in Reddi et al., Adaptive federated optimization, ICLR 2021 (see Section D in the appendix). If the paper gets accepted, please include the updated results with learning rates found from grid search.
> > > >
> > > > I appreciate the authors' rigorousness in investigating this problem and have updated my score from 5 to 6.

---

> > > > > ### Author Response · Authors · 2023-08-21
> > > > >
> > > > > Thanks for raising the score! We appreciate that.  Also, thanks for your suggestion about grid search, roughly speaking, we need to choose smaller learning rates as the value of K decrease as indicated by the convergence analysis, but as K gets smaller, it is harder to find a proper learning rate that leads to sufficient progress. For K=1500, a learning rate of 0.01 leads to 65\% accuracy in 2000 rounds, but for K=500, we find that  for learning rate in {1e-3, 1e-2, 1e-1, 1}, the algorithm won't converge and test accuracy stays around 10\% which is a random guess for MNIST.  Finally, we will perform a more thorough grid search to the hyper-parameters for all algorithms in our final version.

---

### Author Rebuttal · Authors · 2023-08-10

Thanks all the reviewers for their time and effort. Below we provide responses to questions related to the problem formulation, motivation and interpretation of the convergence theorem:

**Q1: Can you formalize the conflict between communication and learning?**

**A**: Formally, the conflicts between communication and learning can be expressed by the following federated optimization problem with constraints: $\underset{x}{\min } h(x) \coloneqq \frac{1}{M}\sum_{m=1}^M L(x; D^{(m)}), \text{s.t. } \text{Comm}(x) \leq C_1, \text{Cap}(x) \leq C_2$. $L(x; D^{(m)})$ denotes the training objective: we fit a model parameterized with $x$ over the dataset $D^{(m)}$, $L$ denotes a loss objective. For the constraints, $\text{Comm}(\cdot)$ denotes the communication constraint, e.g. the bit-rates transferred between the server and the client is upper bounded, while $\text{Cap}(\cdot)$ denotes the computation capacity constraint, e.g. GPU memory size of clients constrains the largest model that can be trained.  For simplicity, we assume both $\text{Comm}(\cdot)$ and  $\text{Cap}(\cdot)$ only depends on the dimension of the parameter $x$. It is straightforward to identify the conflict between communication and learning from this formulation: Firstly, to satisfy the communication constraint, we need to set $x$ in a low-dimensional space, however,  a high-dimensional parameter is desired for optimizing the objective $L(x; D^{(m)})$, furthermore, the capacity limit $C_2$ is decided by the smallest capacity of all clients, i.e. $C_2 = min (C^{(m)}_2, m\in [M])$, where $C^{(m)}_2$ denotes the capacity constraint of client $m$. As a result, clients with larger capacity are under-utilized. One intuitive approach to mitigate the above mentioned issues is by using different sets of parameters to satisfy the two constraints, which is indeed the communication parameter and the learning parameter proposed in the FedSep framework.

**Q2: Why do we need FedSep if several approaches exists for the communication-efficient FL.**

**A**: Compared to existing works in the literature, our FedSep framework (with a bilevel formulation) **provides a novel and different perspective** to tackle the communication-efficiency challenge in FL.
More specifically, existing works often view compression operations (such as quantization and sparsification) as introducing errors into the optimization process, as a result, they either employ error feedback techniques for corrections or leverage information theory tools to determine the optimal encoding strategy within communication constraints. In contrast, FedSep distinguishes between communication parameters and learning parameters. Rather than focusing on compression error analysis as existing approaches, we perform convergence analysis through the lens of bilevel optimization.

As shown in the convergence analysis (Theorem 3.7 and Corollary 3.8), we have that the communication parameter $x$ reaches to a stationary point of the bilevel problem (in Eq.1), and the learning parameter $y^{(m)}$ converges to the stationary point of the local learning problem $f^{(m)}(y)$. In the context of communication-efficient FL, we choose the decode function $g^{(m)}$ to be identical across all clients, therefore, the learning parameter $y^{(m)}$ is also identical across clients. Combining with the convergence result, we show that $y \coloneqq y^{(m)}, m \in [M]$ is a stationary point of the classical FL optimization problem $\frac{1}{M}\sum_{m=1}^M f^{(m)}(y)$. To the best of our knowledge, this is the first convergence result for communication parameter (equivalently, the compressed parameter in the literature).

**Q3: How to interpret the convergence result in Theorem 3.7**

**A**: Firstly, **FedSep shows a Coding-Accuracy trade-off**. As shown in Theorem 3.7 (Corollary 3.8), the encode/decode operation affects the convergence in two folds: firstly, $G_2$, the variance of the encode operator, is a coefficient of the $\frac{1}{\sqrt{T}}$ term; $\tilde{G}$, the bias of encode/decode operations, is an additive term. To control the variance term, we can increase the batch-size $b_x$, and to control the bias term, we can increase the number of decode steps $I_{dec}$ and the number of encode steps $Q$. Since the bias error of both decode and encode operations have linear convergence, we can choose small values of $I_{dec}$ and $Q$ in practice. This is consistent with the Accuracy-Communication trade-off in the compression FL literature. In fact, higher communication rate makes the decode minimization problem more ill-posed (e.g. the decode function in Eq.4 in the manuscript), thus needs more decode/encode steps to reach a given error.


Next, our FedSep matches the standard $O(\epsilon^{-2})$ rate as compression FL [1] for the non-convex function. For the encoding/decoding related dependence: [1] has a linear dependence over the compression error, while our FedSep has a linear dependence over the variance term $G_2$ and an additive dependence of the bias term $\tilde{G}$.

Finally, the convergence rate of FedSep matches that of the federated bilevel optimization method FedNest [2] with a rate of $O(\kappa^5\epsilon^{-2})$. Note that the $\kappa$ is the condition number of the decode function $g^{(m)}$, which is different from the condition number of a strongly-convex objective in classical FL. The convergence bound does not rely on the heterogeneity of client's loss function, as we choose the global learning rate $\eta_g = O(\frac{1}{I})$ to control the error caused by local updates.

**References**

[1] Federated learning with compression: Unified analysis and sharp guarantees, Haddadpour et al., AISTATS 21

[2] Tarzanagh, Davoud Ataee, et al. "Fednest: Federated bilevel, minimax, and compositional optimization." International Conference on Machine Learning. PMLR, 2022.

---

### Decision · Program_Chairs · 2023-09-21

**Decision:**

Accept (poster)

**Comment:**

This work explores a bilevel optimization technique in federated settings that aims to reduce communication costs while maintaining model utility. Reviewers all agreed that the problem formulation was interesting and appreciated the empirical results and convergence guarantees. There were concerns about (1) better motivating/explaining the tension that exists between communication/learning, and (2) more thoroughly explaining how FedSep compares to prior work in personalized and communication-efficient FL. The authors/reviewers had a significant back & forth discussion including discussing some of these ideas in detail; I would recommend the authors carefully incorporate these discussions in their revision.